# Rod nuclear architecture determines contrast transmission of the retina and behavioral sensitivity in mice

Kaushikaram Subramanian[1,2,3], Martin Weigert[1,2,3], Oliver Borsch[4], Heike Petzold[1], Alfonso Garcia-Ulloa[1], Eugene W Myers[1,2,3,5], Marius Ader[4], Irina Solovei[6], Moritz Kreysing[1,2,3]*

[1]Max Planck Institute of Molecular Cell Biology and Genetics, Dresden, Germany; [2]Center for Systems Biology Dresden, Dresden, Germany; [3]Cluster of Excellence, Physics of Life, Technische Universität Dresden, Dresden, Germany; [4]Center for Regenerative Therapies Dresden, Technische Universität Dresden, Dresden, Germany; [5]Department of Computer Science, Technische Universität Dresden, Dresden, Germany; [6]Biozentrum, Ludwig Maximilians Universität, München, Germany

**Abstract** Rod photoreceptors of nocturnal mammals display a striking inversion of nuclear architecture, which has been proposed as an evolutionary adaptation to dark environments. However, the nature of visual benefits and the underlying mechanisms remains unclear. It is widely assumed that improvements in nocturnal vision would depend on maximization of photon capture at the expense of image detail. Here, we show that retinal optical quality improves 2-fold during terminal development, and that this enhancement is caused by nuclear inversion. We further demonstrate that improved retinal contrast transmission, rather than photon-budget or resolution, enhances scotopic contrast sensitivity by 18–27%, and improves motion detection capabilities up to 10-fold in dim environments. Our findings therefore add functional significance to a prominent exception of nuclear organization and establish retinal contrast transmission as a decisive determinant of mammalian visual perception.

*For correspondence:
kreysing@mpi-cbg.de

Competing interests: The authors declare that no competing interests exist.

## Introduction

The structure of the vertebrate retina requires light to pass through multiple cell layers prior to reaching the light-sensitive outer segments of the photoreceptors (*Dowling, 1987*). In nocturnal mammals, the increased density of rod photoreceptor cells demands a thicker (*Němec et al., 2007*; *Peichl, 2005*) rod nuclei-containing outer nuclear layer (ONL). For mice, where rods account for around 80% of all retinal cells (*Hughes et al., 2017*), this layer of photoreceptor nuclei is $55 \pm 5$ μm thick, thus creating an apparent paradox by acting as a more pronounced barrier for projected images prior to their detection (*Figure 1A*). Interestingly, rod nuclei are inverted in nocturnal mammals (*Błaszczak et al., 2014*; *Falk et al., 2019*; *Kreysing et al., 2010*; *Solovei et al., 2009*; *Solovei et al., 2013*) such that heterochromatin is detached from the nuclear envelope and found in the nuclear center, whereas euchromatin that has lower mass density (*Imai et al., 2017*) is re-located to the nuclear periphery. Given that this nuclear inversion is exclusive to nocturnal mammals and correlates with the light-focusing capabilities of isolated nuclei, it was proposed as an evolutionary adaptation to life under low-light conditions (*Błaszczak et al., 2014*; *Kreysing et al., 2010*; *Solovei et al., 2009*). However, the nature of any visual improvements that could arise from nuclear inversion remains unclear.

**Figure 1.** Light scattering by retinal nuclei reduces with chromocenter number during development. (**A**) Longitudinal section showing the path of light through the mouse retina, including the rod nuclei dominated outer nuclear layer (ONL). Ganglion cell layer (GCL), inner nuclear layer (INL) and outer nuclear layer (ONL) and the inner and outer segments (IS and OS). (**B1**) (top) Downregulation of the lamina tether LBR (yellow) enables fusion of mobilized chromocenters and thereby an architectural inversion of mouse rod nuclei. (bottom) FISH images of rod nuclei stained with DAPI (blue)

*Figure 1 continued on next page*

*Figure 1 continued*

showing the dense chromocenters, LINE rich heterochromatin (H4K20me3, magenta) and SINE rich euchromatin (H3K4me3, green) (B2) DAPI section of WT mouse retina in comparison to a Rd1/Cpfl1-KO mouse retina showing the presence of only the inner retina. (B3) Quantification of image transmission shows that the inner retina alone (Rd1/Cpfl1-KO, N = 5) *transmits approximately 50% more image* detail than the full retina (N = 11), suggesting significant image degradation in the thick outer nuclear layer. (C) FACS scattering profiles comparing retinal neurons, cortical neurons and N2a neuroblastoma cells showing lower light scattering properties of retina neurons. (Inset) Volume-specific light scattering is significantly reduced in the retinal cell nuclei. (D, E) FACS scatter plot for isolated retinal nuclei from WT developmental stage week three pup (P25) and adult mice demonstrating stronger large angle scattering by the P25 nuclei. (F) Histogram of side scattering in adult and P25 retina depicting a higher side scattering for the developing retinal nuclei. (G) Sorting of developmentally maturing nuclei according to different side scattering signal. Insets show representative examples of Hoechst stained nuclei in the corresponding sort fractions. The rectangles represent sorting gates for microscopy analysis. (H) Quantification of reduced scattering with chromocenter number is sufficiently explained by a wave optical model of light scattering n = 38 nuclei. (Error bars in (H) show s.d.) Scale bars (A) - 10 µm. (B1), G - 5 µm, (B2) – 50 µm.

The online version of this article includes the following figure supplement(s) for figure 1:

**Figure supplement 1.** Heterochromatin in mouse rod nuclei exhibits unusual dense packing.

**Figure supplement 2.** Reorganization of rod nuclear architecture in the course of postnatal retinal development (A) and in transgenic rods expressing LBR (B, C).

It is widely assumed that high-sensitivity vision depends on optimized photon capture (*Schmucker and Schaeffel, 2004*; *Warrant and Locket, 2004*) and often comes at the expense of image detail (*Cronin et al., 2014*; *Warrant, 1999*). Here, we show that nuclear inversion affects a different metric of vision, namely contrast sensitivity under low-light conditions. In particular, we experimentally show that nuclear inversion improves retinal contrast transmission, rather than photon capture or resolution. Advanced optical modelling and large-angle scattering measurements indicate that this enhanced contrast transfer emerges from previously coarse-grained (*Błaszczak et al., 2014*; *Kreysing et al., 2010*; *Solovei et al., 2009*) changes in nuclear granularity, namely a developmental reduction of chromocenter number (*Figure 1B1*). Moreover, genetic interventions to change chromocenter number in adult mice reduces contrast transmission through the retina, and compromise nocturnal contrast sensitivity accordingly. Our study therefore adds functional significance to nuclear inversion by establishing retinal contrast transmission as a decisive determinant of mammalian vision.

## Results

### Volume-specific light scattering from chromocenters

To test how the presence of densely packed rod nuclei in the light path affects the propagation of light through the retina, we compared transmission of micro-projected stripe images through freshly excised retinae of wild type (WT) (*Figure 1A*, *Figure 1B2* - left image) and Rd1/Cpfl1- KO mice (*Chang et al., 2002*), which lack all photoreceptors including the ONL (*Figure 1B2* - right image). In the absence of photoreceptors and their nuclei, we observed 49% greater imaged detail (cut-off chosen at 50% residual contrast, *Figure 1B3*). Photoreceptor nuclei contain highly compacted and molecularly dense DNA with significant light-scattering potential (*Drezek et al., 2003*; *Marina et al., 2012*; *Mourant et al., 2000*), while photoreceptor segments have been described as image-preserving waveguides (*Enoch, 1961*). These findings suggest that light propagation in the mouse retina is significantly impacted, if not dominated, by the highly abundant rod nuclei of the ONL.

We then asked whether retinal cell somata are optically specialized with distinct light-scattering properties. We compared the light scattering by different cell types using high-throughput FACS (*Feodorova et al., 2015*) measurements. The suspensions of cells or papain-digested retinae were used to measure the cellular light scattering in the far-field using a commercial FACS set up. These measurements revealed that cells isolated from the mouse retina known to typically consist of ~80% rod photoreceptor cells (*Hughes et al., 2017*), scatter substantially less light than neurons of the brain and cultured neuroblastoma cells (*Figure 1C*). This trend is seen for forward-scattered light (measured in a narrow range around 0°) but is even more pronounced for side scattering (measured around 90 degrees, see supplementary methods for details), which reflects subcellular heterogeneity. Using forward scattering as a measure of cell size indicates that side scattering normalized by volume (volume-specific light scattering) is also noticeably lower in retinal cells (*Figure 1C*, inset). This suggests that retinal cells are indeed optically specialized, as they scatter less light for a given

size. This unique property for the rod cells could stem from the unusually dense packing of the heterochromatin in the centre of their nuclei, which notably even excludes free GFP molecules (*Figure 1—figure supplement 1B*).

To determine when the low sideward light scattering characteristic of retinal nuclei emerges, we compared the scattering profile of retinal nuclei in P25 WT pups and WT adult (12 weeks) mice. We found little or no difference between forward light scattering (*Figure 1D–E*), as predicted by earlier models (*Błaszczak et al., 2014*; *Kreysing et al., 2010*; *Nagelberg et al., 2017*). In stark contrast however, sideward scattering, with a strong potential to diminish image contrast, was significantly reduced in adult retinal nuclei compared to the intermediate developmental stage (*Figure 1F*). Quantitative analysis of sorted nuclei from P25 retinae further revealed a monotonic relation between chromocenter number and sideward scattering signal (*Figure 1G*). In particular, those nuclei with the lowest number of chromocenters were found to scatter the least amount of light. In support of this experimental quantification, a wave-optical Mie model of light-scattering by refractive chromocenters closely reproduced the trend of light scattering reduction with chromocenter fusion (*Figure 1H*).

To establish whether rod nuclear inversion is required to cause the developmental reduction in light scattering, we used a transgenic mouse model (TG-LBR) in which heterochromatin remains anchored at the lamina which in turn prevents the complete fusion of chromocenters (*Figure 2A1, A2 and B*, *Figure 1—figure supplement 2*, *Figure 1—figure supplement 1*; *Figure 2—figure supplement 1*) (*Solovei et al., 2013*). FACS experiments of nuclei from TG-LBR retinae, in which > 70% of the nuclei are successfully arrested (*Figure 2—figure supplement 2*), revealed significantly increased light scattering (*Figure 2C*, *Figure 1F*). Specifically, the global maximum of the side-scattering was re-located precisely to the position that is characteristic of nuclei isolated from WT pups at P14, which possess a similar number of chromocenters as inversion arrested nuclei (compare *Figure 2B,C*, *Figure 1F,G*). Because inhibition of chromocenter fusion leads to specific increase in scattering, we conclude that the reduction of light scattering with chromocenter number is causal.

## Improved retinal contrast transmission

Next, we asked how nuclear substructure could affect the optical properties of the ONL. We first approached this via a simulation that built on recent advances in computational optics (*Weigert et al., 2018*). This allowed us to specifically change nuclear architecture, while leaving all other parameters, including the morphology and relative positioning of about 1750 two-photon mapped nuclei, unchanged (*Figure 2D1*, *Videos 1* and *2*) (Supplementary Methods).

These simulations suggest that especially the large-angle scattering (cumulative scattering signal at angles > 30 deg) monotonically decreases when 10 chromocenters successfully fuse into one (*Figure 2D2 and E*). Physically, this effect of reduced scattering can be explained by a reduction of volume-specific scattering for weak scatterers in the size regime slightly above one wavelength of light, similar to scattering reduction techniques proposed for transparent sea animals (*Johnsen, 2012*) (*Figure 2F,G*). Furthermore, a minimal optical ONL model reconstituted from suspended beads of different size but same volume fraction (Supplementary Methods) illustrates how a decreased geometric scattering cross section after fusion leads to reduced scattering-induced veil that helps to prevent contrast losses (*Figure 2H* and inset). Taken together these data suggest that nuclear inversion might serve to preserve contrast in retinal transmitted images.

To experimentally quantify the optical quality of the retina with respect to nuclear architecture, we applied the concept of the modulation transfer function (MTF), a standard way to assess image quality of optical instruments (*Boreman, 2001*). Specifically, MTF indicates how much contrast is maintained in images of increasingly finer sinusoidal stripes (*Figure 3—figure supplement 1B,C*). We therefore devised an automated optical setup (*Figure 3—figure supplement 1A*) that allowed us to project video sequences of demagnified sinusoidal stripe patterns through freshly excised retinae and assess the retinal transmitted images for contrast loss. This custom built set-up mimics the optics of the mouse eye, in particular its f-number (*Schmucker and Schaeffel, 2004*), while circumventing changes of the optical apparatus in-vivo (*Figure 3—figure supplement 1*, Materials and methods).

Strikingly, we found that wildtype retinae improve contrast transmission throughout terminal development, with adult retinae showing consistently elevated MTFs compared to intermediate developmental stages (P14) in which rod nuclei still possess around five chromocenters (*Figure 3A*).

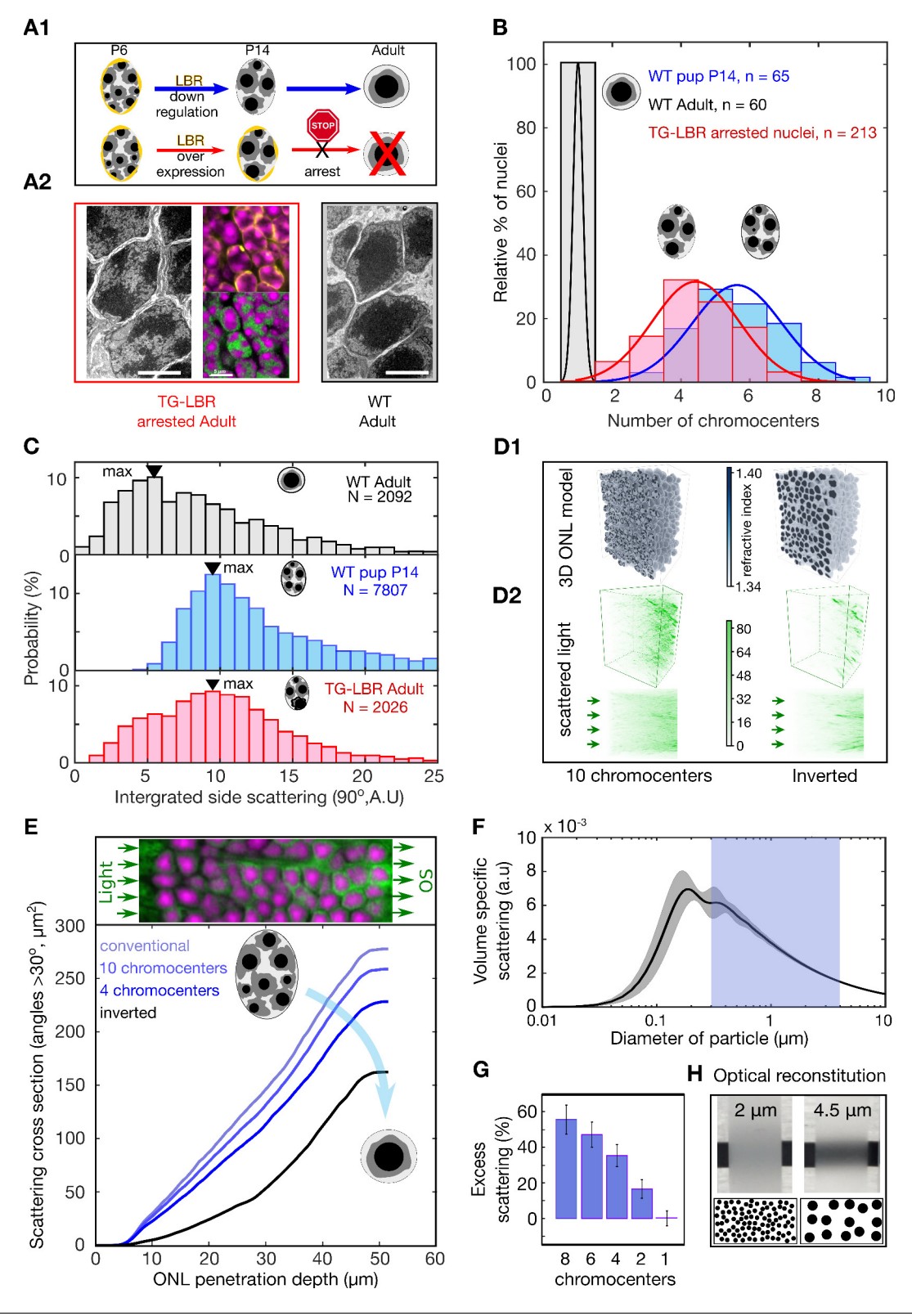

**Figure 2.** Developmental arrest of chromocenter fusion increases light scattering from rod nuclei in measurements and tissue simulations. (**A1**) Schematic of the normal rod nuclear WT development and inversion arrested nuclei by LBR overexpression. (**A2**) EM images illustrating different electron densities in the euchromatic and heterochromatic phase underlying their refractive index (RI) differences (scalebar 5 μm). (mid-top) Immunostaining of overexpressed of LBR tethers (yellow), and high-density heterochromatin (DAPI, magenta). (mid- bottom) Heterochromatic

*Figure 2 continued on next page*

*Figure 2 continued*

chromocenters (DAPI, magenta) and euchromatin (H4K5ac, green) (B) Chromocenter number distribution in LBR overexpressing rod nuclei is drastically different from WT mice, and similar to a developing WT pup (P14). (C) Side scattering assessed by FACS for TG-LBR retina nuclei is higher than that of WT nuclei and comparable to that of a WT P14 nuclei with similar chromocenter numbers. Note the shift of peak value upon LBR overexpression. (D1) 3d RI distribution mapped onto anatomically faithful volumetric ONL images. WT inverted architecture (right, top) and early developmental state (left) (simulation). (D2) (top) Differential simulations of light propagation in the ONL, using same positions and shapes of about 1750 nuclei, but varying chromatin distributions. (bottom) Maximum projection illustrating greater proportions of scattered light (angles > 30 deg) in the ONL with multiple chromo-centered nuclei. (E) Quantitative analysis of this data. (F) Angle weighted volume-specific scattering strength for nuclei models evaluated by Mie scattering theory. (G) Excess scattering occurring in multi-chromocenter nuclei models. (H) Chromocenters scattering reconstituted in an emulsion of silica spheres in glycerol-water mixture. Pictograms reflect accurate number ratio of spheres.

The online version of this article includes the following figure supplement(s) for figure 2:

**Figure supplement 1.** Distribution of chromocenters in nuclei of adult rods transgenitically expressing LBR in comparison to nuclei of P14 WT rods.
**Figure supplement 2.** Transgenic expression of LBR does not influence rod photoreceptor structure.

In contrast to many lens-based optical systems, retinal MTFs have a long tail with non-zero residual contrast despite an initial rapid loss of contrast (a characteristic of scattering-dominated optical systems). The monotonic decay of retina-transmitted contrast indicates scattering-induced veil, rather than a frequency cut-off to be the cause of contrast loss (*Figure 3A,B Figure 3—figure supplement 2A–D*). Collected from >1300 high resolution images, this data reveals that, similar to the lens (*Tkatchenko et al., 2010*), the retina matures towards increasing optical quality during latest developmental stages, with chromocenter fusion as a putative mechanism of veil reduction.

Next, we asked if developmental improvements in contrast transmission of the retina are indeed caused by chromocenter fusion. For this we used mice in which LBR-overexpression largely arrested chromocenter fusion, resulting in an elevated number of chromocenters in the adult animal, similar to P14 WT (*Figure 2A2, B*), without displaying any effect on other morphological characteristics (*Figure 2—figure supplement 2*). Strikingly, repeating MTF measurements on adult retinae of this inversion arrested mouse model (TG-LBR), we find near identical contrast attenuation characteristics as in the developing retina (compare *Figure 3A and B*). Thus, developmental improvements of retinal contrast transmission are indeed mediated by the inversion of rod nuclei. Notably, when we focus on the retinal transmission data within the spatial frequency regime that is relevant for mouse vision (*Alam et al., 2015*; *Prusky et al., 2004*; *Prusky et al., 2000*), (*Figure 3C*) it can readily be seen that the contrast transmission is up to 33% greater in WT compared to TG-LBR at frequencies ~ 0.28 cycles/deg. Equally, the contrast transmission in this behaviorally relevant regime also increases up to 45% in WT adult compared to the pups.

Frequently, the quality of image-forming optical systems is reported as a single parameter value called the Strehl ratio (*Thibos et al., 2004*). Since our image projection setup closely mimics the mouse eye, it allows meaningful comparisons of the Strehl ratios of retinae, by comparing the volumes under MTF curves. With regards to our MTF measurements, we that find the Strehl ratio (computed using the measurements in the spatial frequency range of 0–2 cycles/deg) of a fully developed retina is increased 2.00 ± 0.15 fold compared to that of pups (P14) in which chromocenters fusion was not completed, and similarly 1.91 ± 0.14 fold (ratio of means ± SEM) improved compared to TG-LBR adult retinae (p=3.4055e-08) in which chromocenter fusion was deliberately arrested (*Figure 3D*).

Since the Strehl ratio makes predictions for the peak intensity of a tissue-transmitted point stimulus, we analysed the effect of micro-projecting a point-like stimulus through the mouse retina (diameter here ~3µm, measurement constrained by outer segment spacing). We found that the resulting image at the back of the WT retina had a near two-fold (1.79 ± 0.38, mean ± SD) higher peak intensity compared to the TG-LBR retina (*Figure 3 E1 & E2*, N = 119,

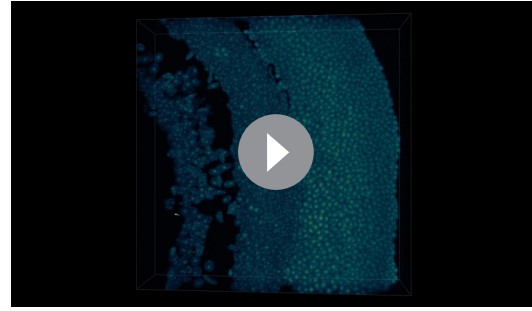

**Video 1.** 2 photon volumetric image of WT mouse retina.
https://elifesciences.org/articles/49542#video1

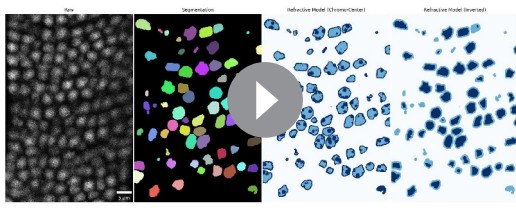

**Video 2.** 3D morphological models of ONL RI distribution used in light propagation simulations. https://elifesciences.org/articles/49542#video2

N = 121, measured in at a total of 6 animals). The measured resolution based on full-width half maximum (FWHM) of the PSF, however, did not show any differences (4.32 ± 2.38 μm, 3.75 ± 2.01 μm, mean ± SD for WT and TG-LBR retinae respectively, *Figure 3 E2*), especially no changes that could physiologically impact acuity, which is known to be significantly lower in mouse. In addition to independently corroborating our MTF measurements, these results emphasize that nuclear inversion enhances contrast transmission through the retina but is unlikely to benefit acuity. From a mechanistic point of view, these measurements indicate that contrast is lost due to the generation of image veil from side scattering, which overcasts attenuated, but otherwise unchanged signals. Accordingly, when comparing the integrated absolute transmission through rhodopsin-bleached retinae in dedicated experiments (Supplementary methods), we found near identical transmission values for WT and inversion arrested retinae (*Figure 3 E3*, $T_{WT}$74 ± 8%, $T_{LBR}$ = 72 ± 5%, mean ± SD), which emphasizes that despite differential image signal, the overall photon arrival at the photoreceptor outer segments, remains unchanged.

An advantage of improved retinal contrast transmission is suggested when following the motion of individual (non-averaged) light stimuli that appear at considerably higher signal-to-noise ratios (*Figure 3F*) at the outer segments level. A putative visual advantage to appropriately scaled real-life examples, such as images of an approaching cat micro-projected through a mouse retina is illustrated in *Figure 3G*. Nuclear inversion results in cat images becoming visible considerably earlier compared to mice that lack nuclear inversion (0.70 vs 0.45 meters, at a given arbitrary noise threshold). These results suggest that nuclear inversion may offer enhanced visual competence that originates from improved contrast preservation in retinal images. More objective and established methodologies to test the impact of nuclear inversion for actual behavior is addressed in the next section.

## Improved contrast sensitivity

To determine whether the improved retinal contrast transmission translates into improved visual perception, we carried out behavioral tests using *Optomotor-reflex* measurements (*Figure 4A*, *Video 3*). Specifically, we used a fully automated mouse tracking and data analysis pipeline (OptoDrum, Striatech, Germany) (*Benkner et al., 2013*) to compare the contrast sensitivities of adult WT mice and those with arrested nuclear architecture (TG-LBR). Firstly, contrast sensitivity assessed by the animal's ability to detect moving stripes, did not differ significantly (p=0.5307, two sample t test) between the two genotypes at photopic light condition (70 Lux – the typical brightness of monitor). Transgenic and WT animals showed comparable visual sensitivity, as quantified by the area under the log-contrast sensitivity curve (AULC, *Figure 4B*, left) (*Villegas et al., 2002*). As nuclear adaptation is strongly correlated with nocturnal lifestyle (*Solovei et al., 2009*; *Solovei et al., 2013*), we adapted this set-up to assess contrast sensitivity under scotopic light conditions. At 20 mLux, which is the range of brightness in moonlight (*Kyba et al., 2017*), we again found comparable responses for coarse stimuli (wide large contrast stripes) suggesting equally functional rod-based vision in TG-LBR and WT mice (*Figure 4B*, right) without noticeable differences in absolute sensitivity. Furthermore, mice deficient of rhodopsin (*Rho-/-*) (*Humphries et al., 1997*; *Jaissle et al., 2001*) confirmed that visual behavior under the displayed scotopic conditions fully relies on the functionality of the rod pathway (*Figure 4—figure supplement 1E*).

When required to detect finer stripes, WT and TG-LBR mice displayed significant differences in their visual performance, specifically in contrast sensitivity (*Figure 4C*). At 20mLux we observed an 18% greater AULC for WT mice compared with TG-LBR mice (p=0.00081). At even lower light intensities (2mLux, comparable to a starry night), the difference in AULC values was even greater (ratio 27%, p=0.0047) albeit at lower absolute sensitivities, which agrees with reported values for WT mice (*Alam et al., 2015*; *Prusky et al., 2000*; *Prusky et al., 2004*). The most significant differences in the

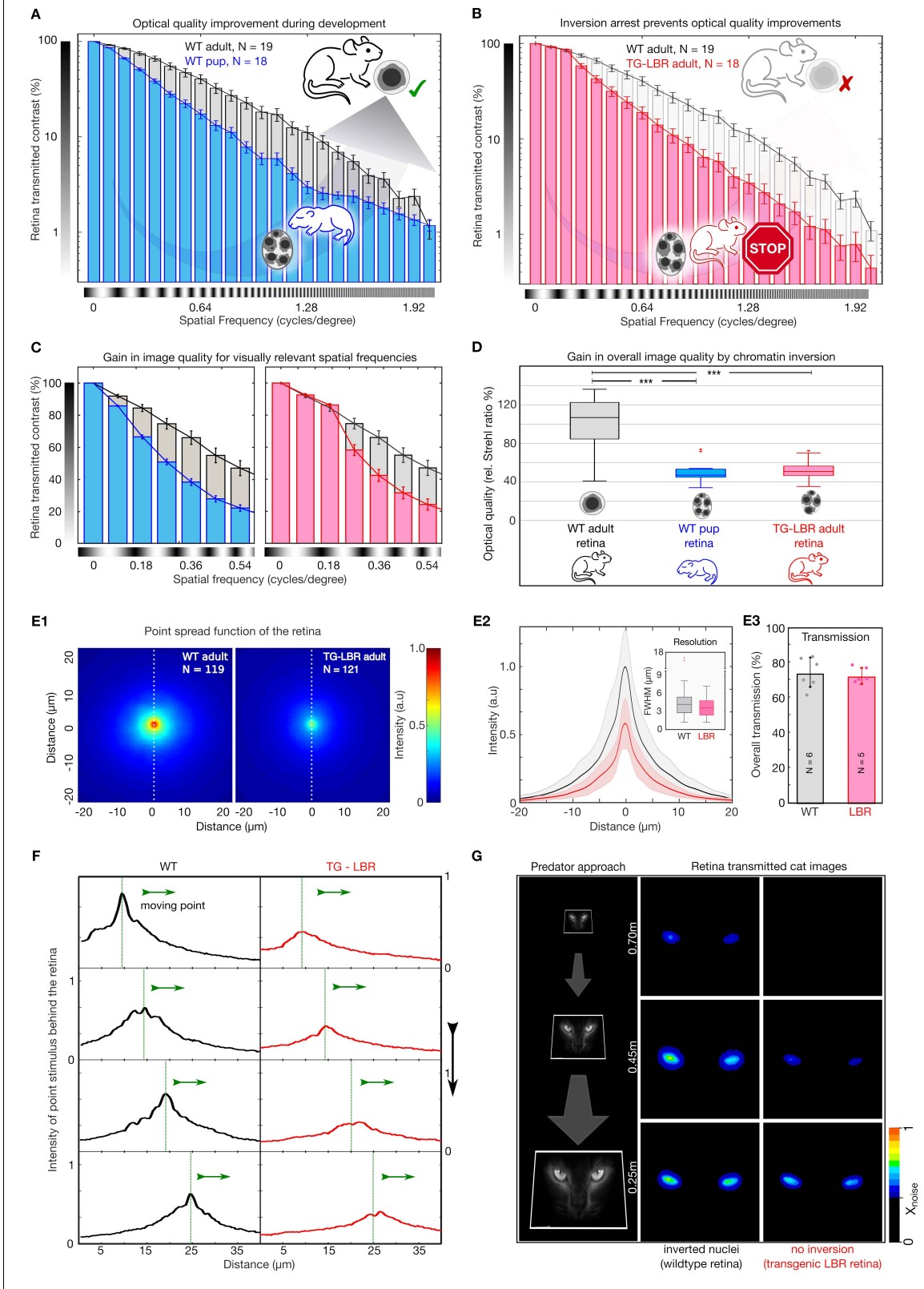

**Figure 3.** Nuclear inversion improves retinal contrast transmission characteristics. (**A**) Retinal contrast transmission increases during developmental stages of nuclear inversion, as experimentally revealed by measurements of retina-transmitted sinusoidal stripe patterns (modulation transfer functions). Developmental stage P12-14 (N = 18), compared to wildtype adult (N = 19 animals), note log scale. (**B**) These improvements in optical quality do not occur in retinae in which rod nuclei are transgenitically arrested in development and maintain 4–5 chromocenters. TG-LBR mouse (N = 18 animals)

*Figure 3 continued on next page*

*Figure 3 continued*

compared to WT reference (N = 19 animals), N = 1950 images in total. Mean +/- 95% CI. (C) Retinal contrast transmission at visually relevant spatial frequencies showcasing on an average ~49% and ~37% better contrast transfer by the WT Adult retina (grey) in comparison to the WT-P14 pup (blue) and TG-LBR Adult (red) respectively. (D) The optical quality improvement of the retina (relative Strehl ratios), as caused by nuclear inversion, is two-fold (p=1.1880e-08 - WT adult vs WT pup, 3.4055e-08 - WT adult vs TG-LBR adult, 0.4761 - TG-LBR adult vs WT pup). (E1) Point spread function (PSF) for WT and LBR adult retinae by projection of 3 µm point light stimuli through the retina, N = 240 measurements in total six retinae. (E2) Intensity quantification along the white dotted line. Shaded region shows ±1 sd. Comparable resolution in transmitted images as assessed by the FWHM of the psf (inset). (E3) Near identical diffuse light transmission by both WT and TG-LBR retinae (n = 2 animals each, mean ± s.d.) (F) Intensity of a moving, retina-transmitted point stimuli for WT (black) and TG-LBR mouse (red). (G) Image-series of a cat approach as seen through the retina of mice, WT and transgenic genotype from various behaviorally relevant distances at the same vision limiting (arbitrarily chosen) signal to noise level. Consistent intensity differences of two or more color shades indicate significantly better predator detection potential for WT mice. Data magnified for clarity.

The online version of this article includes the following figure supplement(s) for figure 3:

**Figure supplement 1.** Simplified schematic of the custom micro-projection setup and the concept of modulation transfer function.
**Figure supplement 2.** Modulation Transfer Function and its relation to light scattering and visual perception.

contrast sensitivity occur above 0.15 cycles/degree (*Figure 4D*). Especially, in the regime close to the visual acuity (0.26–0.30 cycles/degree), WT mice show up to 10 times (p<0.0001) greater positive response rates at intermediate contrasts (*Figure 4E*) compared to mice with inversion arrested rod nuclei. Moreover, at 90–100% contrasts, where WT mice approach a maximum responsiveness, we observed a near 6-fold reduced risk to miss a stimulus for WT compared to TG-LBR mice (false negative rates 11% WT, 59% TG-LBR).

Finally, we asked whether reduced visual sensitivity of mice lacking the inverted nuclear architecture can be sufficiently explained by inferior contrast transmission of the retina. Direct comparison of behavioral sensitivity with the MTF curves showed that vision mostly occurs in regions in which retinal contrast transmission is higher than 50% and substantial differences in MTFs occur. Specifically, the 18–27% difference in contrast sensitivity goes together with a 26% higher Strehl ratio in WT retinae when evaluated in the relevant frequency regime (0–0.36 cycles/degree).

This suggests that at low light levels, contrast sensitivity may be directly limited by contrast transmission through the retina, and that a reduction of contrast sensitivity in mice with non-inverted rod nuclei may be explained by increased contrast losses in the retina. Moreover, we did not observe unexpected side effects from the LBR overexpression, at level of retinal (*Figure 2—figure supplement 2*), ocular or lens anatomy (*Figure 4—figure supplement 2*), and non-limiting rod vision was normal (*Figure 4B* (Left)). Nevertheless, it is clear that the complexity of the eye does not permit an exhaustive comparison of all parameters that could potentially be affected by LBR overexpression, including subtle concentration changes in molecules relevant for phototransduction. So how can one rule out the possibility that the loss of sensitivity in LBR overexpressing mice is due to a loss of image contrast, rather than unspecific side effects?

To show that increased contrast sensitivity in WT mice is due to the increased contrast transmission of the retina, we designed a rescue experiment logically equivalent to rescue experiments that show specificity of molecular interventions. Frequently, one excludes nonspecific side effects of a molecular knock-down by rescuing the phenotype via the addition of the protein of interest (if possible a pathway-specific variant of this protein). To show that sensitivity is lost due to retinal loss of contrast, we performed an optical rescue experiment. For this, we first confirmed that contrast transmission through the inner retina is a linear process, with contrasts at the photoreceptor levels being proportional to contrasts in projected images (*Figure 4—figure supplement 1B,C*). We then adjusted the displayed contrasts in optomotor measurements to pre-compensate for higher contrast losses in the TG-LBR retina while also conserving image intensities. Strikingly, we found that with equal image contrast at the level of the photoreceptor segments, visual competence of LBR mice was rescued and becomes near identical to that of WT mice (*Figure 4F*). Thus, improved retinal contrast transmission is indeed sufficient to explain increased contrast sensitivity in mice.

## Discussion

As an important determinant of fitness, animals evolved a wide range of visual adaptation to see in the dark (*Nilsson, 2009*; *O'Carroll and Warrant, 2017*; *Thomas et al., 2017*; *Warrant and Nilsson,*

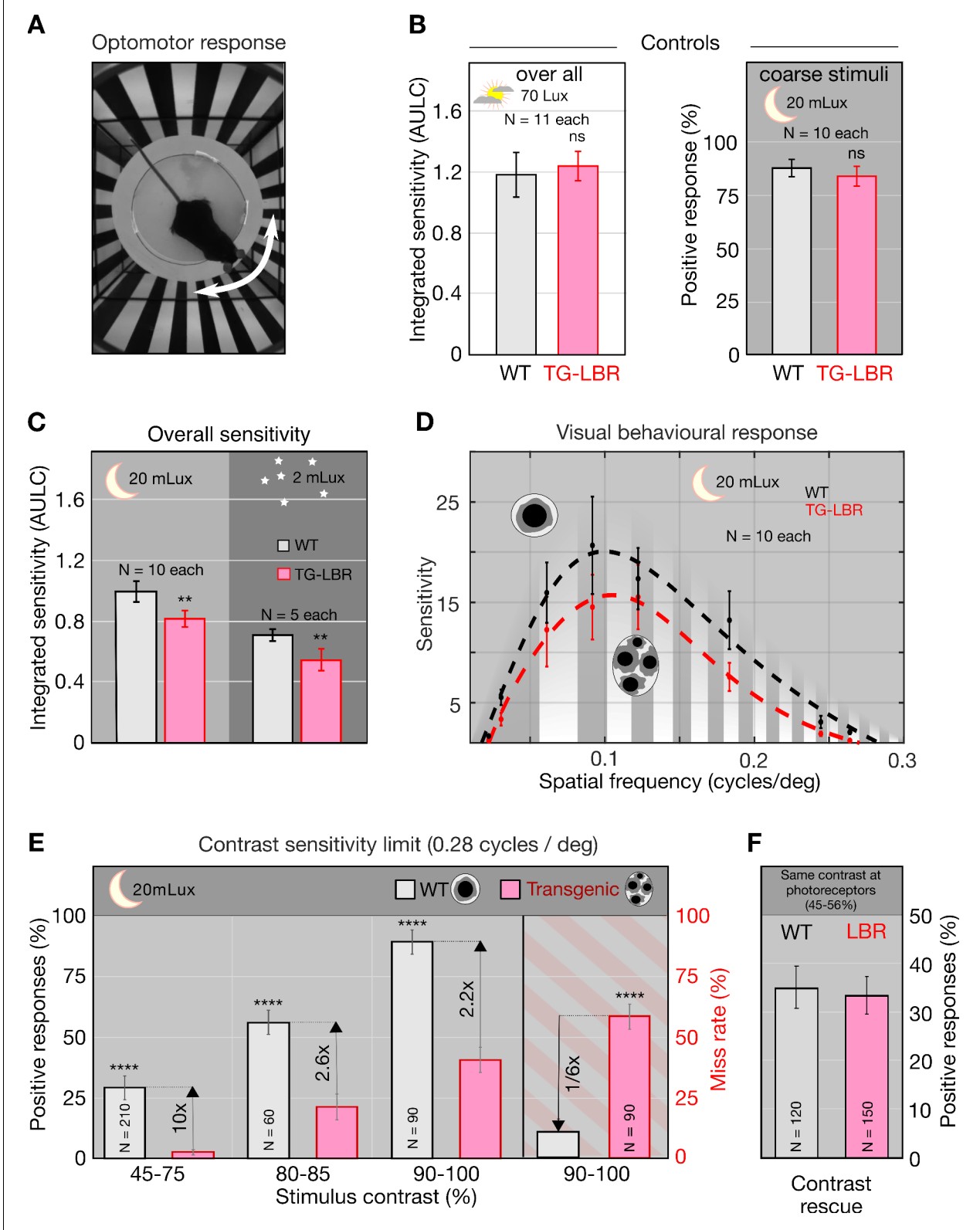

**Figure 4.** Nuclear inversion improves contrast sensitivity in the dark. (**A**) Illustration of the automated optomotor response experiment to assess the visual performance of mice, shown a 0.06 cycles/deg. (**B**) Photopic control condition and scotopic coarse stimulus (0.06cycles/deg) control showing no significant difference between WT and TG-LBR mice (p=0.5307, p=0.2842, t-test, Chi-square test). (**C**) Under scotopic conditions (20 and 2 mLux) the overall sensitivity of the WT mice is 22% and 29% higher than TG-LBR mice (area under log contrast sensitivity curve, AULC) mean+/-95% CI,

*Figure 4 continued on next page*

*Figure 4 continued*

(p=0.00081, 0.0047, two sample t-test). (D) Contrast sensitivity curves evaluated at 20mLux light intensity. Significant differences appear at angular sizes above 0.15 cycles/deg (p=0.038, two sample t-test). (E) Behavior differences are strongest for stimuli close to the visual threshold. Here the mice in possession of the inverted rod nuclei (WT) possess an up to 10 times higher sensitivity at intermediate contrasts (29% vs 3% correct response in 45–75% contrast range), and a six times reduced risk to miss a motion stimulus at high contrasts (10 vs 59% failure in detection, 0.26–0.3 cycles/deg), (p<0.0001, Chi-square test) mean ± s.d. (F) Rescue experiment demonstrating sufficiency of improved retinal contrast transmission to explain improved sensitivity. Adjusting the level of contrast at the photoreceptor level (by pre-compensation of differential contrast loss) restores sensitivity of TG-LBR mice. N indicates number of individual trials of 10 animals together for each mouse type.

The online version of this article includes the following figure supplement(s) for figure 4:

**Figure supplement 1.** Retina transmitted contrast directly impacts visual behavior.
**Figure supplement 2.** Ocular parameters of WT and TG-LBR mice are comparable.
**Figure supplement 3.** Model of nuclear adaptation enhancing nocturnal vision.

*2006*; *Warrant, 2017*). Nocturnal vision is known to rely on highly efficient light capture, both at the level of the lens and photoreceptor outer segments, and often compromises spatio-temporal resolution by summation strategies of neuronal readout (*Warrant, 1999*; *Warrant, 2017*). Here we established nuclear inversion as a complementary strategy to maximize sensitivity under low light conditions. Centrally, we show that it is the direction into which light is scattered inside retinal tissue that translates into differential contrast sensitivity. Specifically, we find that the forward scattering characteristic of inverted nuclei (*Kreysing et al., 2010*; *Solovei et al., 2009*) mainly suppresses light scattering by nuclear substructure towards large angles, thus preventing image veil and contrast reduction resulting from it.

As mammalian eyes are evolutionarily multi-constrained systems, one could ask if nuclear inversion might also serve other functions beyond the improved contrast sensitivity that we have showed. Slightly reduced thickness of the ONL might translate into more efficient diffusion of nutrients, waste and signals. Similarly, the unusually large fraction of hetero-chromatin in rod photoreceptor cells (*Wang et al., 2018*), which might enable the small nuclear volume and/or more efficient fate-specific gene silencing (*Becker et al., 2017*; *Hiler et al., 2015*; *Mattar et al., 2018*; *Wang et al., 2018*), might hypothetically lead to architectural problems for the nucleus which could be circumvented by nuclear inversion. As an example, chromatin distribution is known to have the potential to modulate the mechanical properties of the nucleus (*Kirby and Lammerding, 2018*; *Miroshnikova et al., 2017*; *Stephens et al., 2017*) and LBR downregulation might even be required for shape changes enabling efficient packing of nuclei (*Stephens et al., 2019*). Lastly, although the size of the PSF is beyond the acuity limit of nocturnal vision, the increase in intensity of a point stimulus at the photoreceptor level could aid a thresholded or otherwise non-linear readout of rod cells, a long-standing hypothesis in the field of visual neuroscience (*Barlow, 1956*; *Field and Rieke, 2002*; *Nelson, 2017*) which was substantiated by the use of quantum-based single-photon sources (*Tinsley et al., 2016*).

While these additional functions of nuclear inversion currently remain speculations, it is worth reflecting about the relevance of the visual benefits demonstrated here for enhancing animal vision in general. Since our reported mechanism involves improvements in retinal image contrast rather than notable changes in photon transmission that could impact absolute sensitivity (*Banks et al., 2015*; *Cronin et al., 2014*; *Nilsson, 2009*; *Warrant, 1999*), one might ask why nuclear inversion as an adaptation is exclusive to nocturnal mammals. Wouldn't improvements in retinal image contrast not also be beneficial for diurnal mammals? Firstly, the larger spacing of photoreceptor segments in the diurnal retina significantly reduces ONL thickness

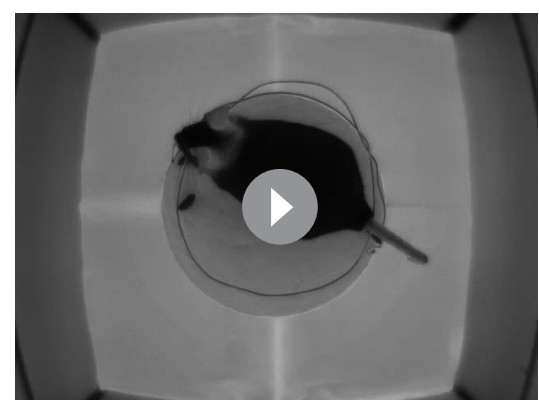

**Video 3.** Behaving mouse in an Optomotor response set up.
https://elifesciences.org/articles/49542#video3

(*Solovei et al., 2009*; *Sterling and Laughlin, 2015*; *Werner and Chalupa, 2004*; *Williams and Moody, 2004*) and thereby the risk of scattering induced veil and loss of image contrast. Furthermore, as is well known from photography, shot-noise that accounts for image granularity (*Barlow, 1956*; *de Vries, 1943*; *Rose, 1948*) becomes less of a problem with increasing light levels. Million-fold higher light intensities during the day imply a higher safety margin from this noise floor (*Warrant, 1999*), (*Figure 4—figure supplement 1F,G*), as required for neural mechanisms of contrast enhancement to function (*Artal et al., 2004*; *Flevaris and Murray, 2015*; *Hess et al., 1998*; *Shevell et al., 1992*). Such compensatory mechanisms are also likely to explain why no behavioral differences are observed at elevated intensities and why augmented vision becomes pronounced only at low light levels. Last, but not least, our measurements show that, although nuclear inversion improves retinal contrast transmission via reduced image veil, resolution, the limiting factor for high acuity diurnal vision, remains largely unaffected. Besides a reduced need for inverted photoceptor nuclei in diurnal mammals, reduced efficiency of canonical DNA repair mechanisms (*Frohns et al., 2014*) in highly condensed chromocenters, could mean a significant disadvantage in the diurnal retina and susceptibility to stress and degeneration (*Boudard et al., 2011*; *Dyer, 2016*), could also mean a significantly higher cost for inverted nuclei in diurnal species, as their retinae are intrinsically strongly exposed to high-energy, ultra-violet photons.

In conclusion, we showed that rod nuclear inversion is necessary and sufficient to explain optically enhanced contrast sensitivity in mice (*Figure 4—figure supplement 3*). Our work thereby adds functional significance to a prominent exception of nuclear organization and establishes retinal contrast transmission as a new determinant of mammalian fitness.

# Materials and methods

## Key resources table

| Reagent type (species) or resource | Designation | Source or reference | Identifiers | Additional information |
|---|---|---|---|---|
| Strain, strain background (*M. musculus*) | C57BL/6NRj, (WT) | Janvier Labs | | Colony maintained at biomedical facility of MPI-CBG |
| Strain, strain background (*M. musculus*) | Tg(Nrl-EGFP) | Kind Gift from Jung-Woong Kim (Anand Swaroop laboratory, Ophthalmology and Visual Sciences, University of Michigan, Ann Arbor). | | (*Akimoto et al., 2006*) |
| Strain, strain background (*M. musculus*) | ROSA26-eGFP-DTA | Kind gift from Dr. Dieter Saur Klinikum rechts der Isar, Technische Universität München | | (*Ivanova et al., 2005*) |
| Strain, strain background (*M. musculus*) | TG-LBR (Nrl-*Lbr*) | This paper, Dr. Irina Solovei, LMU Munich (*Solovei et al., 2013*) | biomedical facility of MPI-CBG | Materials and methods Tissue preparation for optical characterization and Flow cytometry |
| Strain, strain background (*M. musculus*) | Rd1/Cpfl1-KO | Ader Lab, CRTD Dresden, TU Dresden | Animal facility of CRTD | Materials and methods |
| Strain, strain background (*M. musculus*) | Rho-/- | (*Humphries et al., 1997*; *Jaissle et al., 2001*) | Animal facility of CRTD | Materials and methods |
| Cell line (*M. musculus*) | Neuro-2a (Neuroblast cells) | DSMZ | ACC-148; RRID: CVCL_0470 | Cell line maintained as per ATCC recommendations |

*Continued on next page*

*Continued*

| Reagent type (species) or resource | Designation | Source or reference | Identifiers | Additional information |
|---|---|---|---|---|
| Biological sample (*M. musculus*) | Retina | This paper | biomedical facility of MPI-CBG, Animal facility of CRTD | Materials and methods Tissue preparation for optical characterization and Flow cytometry |
| Biological sample (*M. musculus*) | Brain sections | This paper | biomedical facility of MPI-CBG | Materials and methods Flow cytometry |
| Antibody | anti-lamin B (Goat, polyclonal) | Santa Cruz | SC-6217, RRID: AB_648158 | IF (1:50) |
| Antibody | anti-LBR (Guinea pig, polyclonal) | Kind gift from Dr.H.Herrmann (DKFZ, Heidelberg) | | IF (1:50) |
| Antibody | anti- H4K5ac (Mouse monoclonal) | Kind gift from Dr.H.Kimura (Tokyo Institute of Technology, Yokohama) | Clone 4A7 | IF (1:100) |
| Sequence-based reagent | FISH Probes | This paper, Refer to methods for primer sequences. | PCR primers | (*Solovei, 2010*; *Solovei et al., 2007*). |
| Commercial assay or kit | Papain dissociation system kit | Worthington Biochemical Corporation | PDS LK003150 | |
| Software, algorithm | Mie calculations | MATLAB Script | omlc-mie | (*Mätzler, 2002*) |
| Software, algorithm | Calculation of MTF | MATLAB | https://de.mathworks.com/products/matlab.html | Version 2017b, 2018b, |
| Software, algorithm | Retinal light propagation | biobeam | biobeam | (*Weigert et al., 2018*) |
| Software, algorithm | SPSS | IBM, SPSS | ibm-spss | Version 25 |
| Software, algorithm | Optodrum | Striatech GmbH | Striatech | |
| Other | Alexa555 | Invitrogen | A31570; RRID: AB_2536180 | Fluorescent dyes |
| Other | Alexa 488 | Invitrogen | A21202; RRID: AB_141607 | Fluorescent dyes |
| Other | Hoechst | Thermo Scientific | 33342 | Fluorescent dyes |
| Other | Vectashield | Vector Laboratories, Inc, USA | Cat. No. H-1000–10 | Antifade media |
| Other | Aqua Poly-Mount | Polysciences, Inc, USA | Cat. No. 18606–20 | Antifade media |
| Other | FACS tubes | Corning Inc, USA | REF 352054 | Falcon round bottom polystyrene |
| Other | Research Beads | BD Biosciences | 655050 | BD FACSDiva CS and T |
| Other | Silica beads | Whitehouse Scientific | MSS002, MSS004a | |
| Other | ND-1.2 filter | Rosco Laboratories Inc | e-color+ #299, | |

## Retina sampling and preparation of cryo-sections

Wild type retinas were sampled from C57/BL6 mice. Eye balls of Nrl-GFP mice (*Akimoto et al., 2006*) were kindly provided by Jung-Woong Kim (Anand Swaroop laboratory, Ophthalmology and Visual Sciences, University of Michigan, Ann Arbor). Tissues from ROSA26-eGFP-DTA mice (*Ivanova et al., 2005*) were kindly provided by Dieter Saur (Klinikum rechts der Isar, Technische Universität München). The Rd1/Cpfl1-KO mice were maintained in the Animal facility of the CRTD, Dresden. Preparation of retina cryosections was performed according to protocol described earlier

(*Eberhart et al., 2013*; *Eberhart et al., 2012*; *Solovei, 2010*). The enucleated eye balls were shortly washed with EtOH, punched with gauge 23 needle in the equatorial plane and fixed with 4% para-formaldehyde (PFA) (Carl Roth GmbH, Germany) in phosphate-buffered saline (PBS) solution for 3 hr. After fixation, samples were washed with PBS 3x 1 hr each, incubated in 10%, 20% and 30% sucrose in PBS at 4°C for 30 min in each concentration and left in 30% sucrose for overnight. The eyeballs were cut equatorially to remove the anterior parts, including cornea, lens and the vitreous, and eye cups were placed in a mold (Peel-A-Way Disposable Embedding Molds, Polysciences Inc) filled with tissue freezing medium (Jung tissue freezing medium, Leica Microsystems). Frozen blocks were prepared by either immersion of molds with tissues in freezing medium in a 100% ethanol bath precooled to −80°C, or by placing into a container filled with precooled to −70°C 2-methylbutane. After freezing, blocks were transferred to dry ice and then stored at −80°C. Cryosections with thickness of 14–20 μm were prepared using Leica Cryostat (Leica Microsystems) and collected on Super-Frost (Super Frost Ultra Plus, Roth, Germany) or StarFrost microscopic slides (StarFrost, Kisker Biotech GmbH and Co). After cutting, sections were immediately frozen and stored in at −80°C until use.

## Immunostaining

Immunostaining was performed according to the protocol described in detail earlier (*Eberhart et al., 2012*; *Eberhart et al., 2013*). Prior to immunostaining, slides with cryosections were removed from −80°C freezer, allowed to thaw and dry at room temperature (RT) for 30 min and then re-hydrated in 10 mM sodium citrate buffer for 5 min. For the antigen retrieval, slides were transferred to a preheated to +80°C 10 mM sodium citrate buffer either for 5 min (H4K5ac) or for 25 min (lamin B and LBR staining). After brief rinsing in PBS at RT, slides were incubated with 0.5% Triton X100/PBS for 1 hr, and once more rinsed in PBS before application of antibodies. Primary and secondary antibodies were diluted in blocking solution [PBS with 0.1% Triton-X100, 1% bovine serum albumin (ICN Biomedicals GmbH) and 0.1% Saponin (SERVA)]. Incubation with antibodies was performed for 12–14 hr under glass chambers in humid dark boxes (*Solovei, 2010*; *Solovei et al., 2007*). Washings after incubation with antibodies were performed with PBS/0.05%Triton X-100, 3x 30 min at 37°C. Primary antibodies included anti-lamin B (Santa Cruz, SC-6217), anti-LBR (lamin B receptor; kindly donated by Harald Herrmann, German Cancer Research Center, Heidelberg), anti-H4K5ac (kindly donated by Hiroshi Kimura, Tokyo Institute of Technology, Yokohama). Secondary antibodies were anti-mouse IgG conjugated to Alexa555 (A31570, Invitrogen) and Alexa488 (A21202, Invitrogen). Nuclei were counterstained with DAPI or Hoechst added to the secondary antibody solution. After staining, the sections were mounted under a coverslip with Vectashield (Vector Laboratories, Inc, Burlingame, CA, USA) or Aqua Poly-Mount (Polysciences, Inc, USA) antifade media and sealed with nail polish.

For microscopic analysis of FACS sorted retinal nuclei, sorted nuclei were fixed with 4% PFA in PBS for 10 mins, stained with Hoechst 33342, washed 2x with PBS and mounted on slides under coverslips in antifade medium (see below). The imaging was performed on a confocal microscopy (Zeiss LSM 700 inverted) using a Zeiss 64x 1.4 oil objective.

## FISH

FISH on cryosections was performed as described earlier (*Solovei, 2010*; *Solovei et al., 2007*). Probes for LINE, B1 and major satellite repeat (MSR) were generated by PCR using the following primers:

5'-GCCTCAGAACTGAACAAAGA and 5'-GCTCATAATGTTGTTCCACCT for LINE1;
5'-CACGCCTGTAATCCCAGC and 5'-AGACAGGGTTTCTCTGTA for B1;
5'-GCGAGAAAACTGAAAATCAC and 5'-TCAAGTCGTCAAGTGGATG for MSR.

Probes were dissolved in hybridization mixture (50% formamide, 10% dextran sulfate, 1xSSC) at a concentration of 10–20 ng/μl and hybridized to sections of mouse retina for 2 days. Post-hybridization washes included 2xSSC at +37°C (3x 30 min) and 0.1xSSC at +61°C (10 min). Sections were counterstained with DAPI and mounted as after immunostaining (see above).

## Microscopy and image analysis

Single optical sections or stacks of optical sections were collected using either Zeiss LSM 700 or Leica TCS SP5 confocal microscopes equipped with Plan Apo 63x/1.4 NA oil immersion objective and lasers with excitation lines 405, 488, and 561 nm. Dedicated plugins in the ImageJ (*Schindelin et al., 2012*) program were used to compensate for axial chromatic shift between fluorochromes in confocal stacks, to create RGB stacks/images, and to arrange optical sections into galleries (*Ronneberger et al., 2008*)

To estimate the proportion of rods expressing LBR in retinas from TG-LBR mice, four stained cryosections from two homozygous mice were imaged. Not less than 12 image fields with pixel size of 100 nm were collected through each section. Scoring of LBR-positive and negative rods was performed in ImageJ using Cell Counter plugin. Number of chromocenters in TG-LBR rods and P14 WT pups was estimated in confocal stacks through retinas after FISH with major satellite repeat and lamin B immunostaining. Scoring of chromocenters in 210 and 65 nuclei of transgenic and P14 rods, respectively, was performed manually using ImageJ.

## Electron microscopy

For electron microscopy, eyes of adult WT and TG-LBR mice were fixed by cardiac perfusion with a mixture of 2% paraformaldehyde and 2.5% glutaraldehyde in 0.1M cacodilate buffer for 5 min. After eye enucleation, the eye-balls were further fixed in the same fixative for 1 hr and then postfixed with $OsO_4$ in cacodilate buffer for 1.5 hr. Ultra-thin sections were stained with uranyl acetate and Reynolds lead citrate. Images were recorded with a megaview III camera (SIS) attached to a Philips EM 208 transmission electron microscope (FEI) operated at 70 keV.

## Flow cytometry

FACS scattering analysis of retinal cells and sorts according to light scattering profiles were performed based on previously published methodologies (*Feodorova et al., 2015*). The Flow cytometric analysis was performed using FACS Aria Fusion (BD Biosciences) equipped with 488 nm laser and a 70 μm nozzle. For performance tracking and to ensure stability of the scattering signal, calibration beads from BD biosciences (BD FACSDiva CS and T Research Beads, 655050) were used. The scattering signal height vs width was then used to gate for singlet cell populations. Cell aggregates and debris were excluded for the data analysis. The papain dissociation system kit from Worthington Biochemical Corporation was used to digest the tissues. All the solutions for the digestion were prepared according to the manufacturer's recommendation. The retinae from adult mice were gently and quickly isolated from enucleated fresh unfixed mouse eyeballs. 250 μl Papain digestion solution contained in a 2 ml Eppendorf tube was equilibrated (for 15–20 min) in 5% CO2. Two to four retinae were transferred to the equilibrated solution and incubated in a thermomixer at 37°C at 700 rpm. The tubes were periodically checked by visual inspection to ensure proper dissolution of the tissue. After 15–20 mins of incubation, the digest was added to a tube containing 15 μl DNAase-EBSS. The mixture was then mechanically agitated by pipetting the solution up and down 10 times with a 1 ml pipette until no tissue pieces are visible. After mechanical dissociation add to the mixture 400 μl ovomucoid-EBSS (10% v/v) a papain inhibitor to arrest further chemical dissociation.

For retinae from young mice (P14 and P25) digestion times were adapted from 20 to 10 min to compensate for a faster dissociation. For brain cortical cells, above described digestion was preceded by vibratome slicing of freshly obtained mouse brains.

Neuro-2a cells were trypsinized (0.05% Trypsin-EDTA, Thermo Fisher Scientific) and washed once with cold PBS. The relevant details of the cell lines used have been included in the Key resource table. The Neuro-2a (Mouse neuroblastoma) cells were obtained as frozen vials supplied and quality controlled by DSMZ, Germany (ACC-148; RRID: CVCL_0470). Cells were purchased in the year 2010, but were not long term cultured since then. Instead they had been stored in liquid nitrogen for the predominant amount of time and were only thawed days before the sorting experiment. In general cell culture facilities are regularly checked for bacterial infections, including mycoplasma infections, and there is no evidence suggesting an infection of the cells. Notably, cells were not recultured after FACS characterisation. In all cases the samples were filtered into Falcon round bottom polystyrene FACS tubes (Corning Inc, USA) using a 40 μm mesh cell strainer (FALCON, Corning Inc, USA) prior to FACS analysis.

For the calculation of the volume specific scattering, the side scattering area was normalized by volume of nuclei by taking the forward scattering area as a measure for size. Volume-specific scattering thus refers to the light scattering normalized by the amount of material, used to compare the light scattering by a material of given volume/mass but different size distribution.

## Mie models of nuclei

The scattering intensity calculations for the multi-chromocenter-nuclei depicted in *Figure 1H*, were performed using Mie scattering models of spheres in a refractive index contrast of 2% (*Kreysing et al., 2010*), the reported contrast of refraction between heterochromatin and euchromatin. Mie calculations were implemented via a MATLAB script (*Mätzler, 2002*) that can be downloaded at the following link - https://omlc.org/software/mie/. The relevant parameters used were m_euchromatin/medium = 1.02, m_heterochromatin = 1.04 (*Kreysing et al., 2010*) which are refractive index of the euchromatin/medium and heterochromatin/particles respectively. The wavelength used was 500 nm and volume fraction vf = 0.3351. The diameter of particles used were in the range 0.9–4 µm. Relative scattering efficiencies for packed scatterers represented in *Figure 3—figure supplement 2 (E)* were calculated based on dependent scattering models (*Twersky, 1978*).

## Micro projection setup

Ex-vivo retinal transmission measurements were carried out using a dedicated custom built, automated optical setup. This micro-projection setup (*Figure 3—figure supplement 1 (A)*) consisted of two distinct optical paths, one containing projection optics (functioned akin to the optics of the eye) that relayed images displayed by the projector LCD on to the image plane of the projection objective lens, and a second that recorded the retina transmitted images. The light source used (ML505L3, Thorlabs) had a spectrum close to that of the sensitivity of the rods ~ 510 nm. The objective lens (NA = 0.45, NPL Fluotar, Leitz, Germany) was chosen to closely match the f# number of the mouse eye (f#~1; *Geng et al., 2011*), with an added option to narrow the incident angular spectrum for absolute transmission measurements. The projected image on the retina was then collected via an imaging/efflux objective (Olympus U PlanApo 20x 0.75/inf corr) and recorded on an Andor Zyla-5.5 sCMOS camera.

## Calculation of MTF

To quantify Modulation Transfer Functions (MTF) spatially extended sinusoidal stripe patterns of different spatial frequency were micro-projected using a custom optical setup (*Figure 3—figure supplement 1A*) and transmitted images were recorded. The MTF was calculated as the ratio of the contrast in the transmitted image and the projected image. With a customized digital projector setup, the implementation of the sinusoidal stripe projection became a straightforward analysis of the optical property for wide retinal regions (~625 µm x ~ 750 µm) (Figures. *Figure 3—figure supplement 1A,C*), *Figure 3—figure supplement 2A-D*). The projection of spatially extended images that display many periods is however also key to capture image veil, since scattering at large angles may reduce contrast not locally (from one peak into the neighboring minimum) but across multiple stripe periods of the test image. In industries MTF is predominantly used to assess various optical systems such as lens, cameras, displays etc. (*Williams and Becklund, 1989*). An advantage of the MTF approach over any spatial domain approach (i.e. PSF analysis) is that overall performance of a system with optical components in series can be conveniently described as a product of the MTFs of the individual components (*Boreman, 2001*). In particular, MTF describes the frequency domain performance of an optical system as a ratio of the contrasts in the output image to the input object as given below,

$$\text{Contrast} = \frac{\text{I}_{\text{max}} - \text{I}_{\text{min}}}{\text{I}_{\text{max}} + \text{I}_{\text{min}}} - \frac{\text{I}_{0,\text{max}} - \text{I}_{0,\text{min}}}{\text{I}_{0,\text{max}} + \text{I}_{0,\text{min}}}; \ \text{MTF}(\xi) = \frac{\text{Contrast}_{\text{image}}(\xi)}{\text{Contrast}_{\text{object}}(\xi)}$$

where, I is the image intensity and $\xi$ is the spatial frequency (number of stripes per unit distance). Practically, the raw images of the stripe patterns were first flat field corrected using Fiji (*Schindelin et al., 2012*) to ensure no global changes in contrasts affected further calculations. Each image was then processed using built in functions in MATLAB by taking an average along the direction orthogonal to the contrast modulation. The resulting one-dimensional sinusoidal intensity

pattern was fit to a sine wave to extract $I_{max}$ and $I_{min}$. (*Figure 3—figure supplement 2A-D*. Subsequently, the MTF was calculated according to the above formula. The MTF of the retina was then obtained by normalizing the measured MTF against the MTF of the optical setup alone. The differential readout of the transmitted image through the inversion arrested TG-LBR retina allows an explicit understanding of the optical impact of the inner retina and the outer nuclear layer architecture in relation to other ocular constituents, such as the lens and the reported optical properties of mouse eye in in vivo studies (*Geng et al., 2011*; *de la Cera et al., 2006*; *van Oterendorp et al., 2011*). As for the photoreceptors outer segments, their impact is minimal as they act as waveguides as described in previous ex vivo studies (*Ohzu et al., 1972*).Such an effect is also verified by our simulations.

## Calculation of Strehl ratio

Strehl ratio (SR) is a commonly used single number estimate of the optical performance of a system that can also be used to evaluate the optical performance of ocular components (*Marsack et al., 2004*; *Thibos et al., 2003*). The SR in the spatial domain is formally defined to be the ratio of the peak intensities of a PSF to that of a diffraction limited PSF (*Strehl, 1895*). In terms of the frequency domain analysis, one can more accurately calculate the SR by taking the volume under the Optical Transfer Function, albeit for systems with negligible phase transfer properties (as planar tissues), the volume under the MTF suffices to calculate the SR. This way the SR was calculated for each biologically independent sample by taking area under the frequency weighted MTF curve along the spatial frequency (*Figure 3—figure supplement 2F*).

## PSF measurements

The point spread function (PSF) measurements were carried out using a 40 µm pinhole (P40H, Thorlabs) acting as a point light source, such that the demagnified point projected on the retina was of the size about 3 µm. Raw images were corrected for background by subtraction of a dark frame in FIJI. Resulting images were normalized with respect to the integral intensity in the field of view (~80 µm by ~ 80 µm), and the central region with an ROI of 40 µm by 40 µm was cropped, averaged and displayed in false color.

## Diffuse transmission measurements

The measurements were carried out with the micro projection setup above such that a point source was projected through an effective NA of 0.05 on to the retina with a final size of ~30 µm diameter. The transmitted light was collected using an Olympus UPlanSApo 40x 1.25 NA silicone immersion objective lens and recorded on the camera. The fractional transmission of the samples was then calculated, after subtraction of a dark frame reference, based on the integrated intensity in the entire field of view compared to the intensity without the sample in place.

## Hiding power

The angular-weighted integrated scattering intensity is also known as hiding power. Specifically, hiding power is represented as the product of the efficiency of scattering ($Q_{sca}$) and the directional weightage component, otherwise known as the anisotropy factor (g) (*Johnsen, 2012*). The theoretical calculations based on Mie models presented in Figures(1H, 2 F-G) were done using a MATLAB script reported by *Mätzler (2002)* that can be downloaded at the following link - https://omlc.org/software/mie/. The relevant parameters used were m_euchromatin/medium = 1.02, m_heterochromatin = 1.04 (*Kreysing et al., 2010*) which are refractive index of the euchromatin/medium and heterochromatin/particles respectively. The wavelength used was 500 nm and volume fraction vf = 0.3351. The diameter of particles used were in the range 0.92–4 µm.

## Optical reconstitution

Equal amounts (by weight) of silica beads of diameter 2 µm (MSS002) and 4.5 µm (MSS004a, lot obtained from Whitehouse Scientific) of RI 1.48 were dispersed in separate cuvettes containing glycerol-water mixture (RI = 1.43), the larger beads closely resembled the size of heterochromatin mass after chromocenter fusion, and the smaller beads corresponded to a ~ 12 chromocenter case nucleus

(at a conserved total volume/mass). An edge was imaged through the two dispersions using a commercial mobile phone camera with a LED white lights acting as a light source.

## Tissue preparation for optical characterization

Animals were sacrificed by cervical dislocation, and one eyeball immediately removed and opened in fresh environmentally oxygenated PBS. Next, the anterior of the eye, including the cornea and the lens, was fully removed. The retina was gently detached from the choroid, the optic nerve clipped and pulled out from the posterior cup. The retinal cup was placed on a 22 × 60 mm coverslip. Special attention was given to remove any residual vitreous humour sticking to the retina. While the retina remained floated in PBS radial incisions were made and the retinas were flattened on the coverslip by aspiring tiny amounts of the PBS. An appropriately flattened retina was mounted under a smaller coverslip in PBS. A 255 µm spacer was placed between the two coverslips under a stereomicroscope to prevent squeezing of the retina. Preparation were typically achieved in 2 min, and no retina was considered for measurement with a preparation time of more than 5 min. Optical measurements were done in an automated fashion with results in adult WT mice comparable to double pass experiments in vivo (*Artal et al., 1998*).

## Behavioral assessment - Optomotor response

Visual behavioral response was assessed using a fully automated, monitor based optomotor drum setup obtained from Striatech (Striatech GmbH, Tübingen, Germany) and the experiments were conducted at the CRTD, TU Dresden, Germany. The optomotor setup was a closed box with four digital displays to simulate a rotating cylinder of stripe patterns. An opening above allowed the view of the animal via a camera. An independent computer-controlled software was used to track the mice on the platform. The presentation of the pattern and scoring of the movement tracking performance was done through a proprietary software program. Software details can be found in *Benkner et al. (2013)*.

Age (5–6 months old) and gender matched mice from Wild type and TG-LBR (transgenic) mice were used for comparison of the behavior. The tests were performed under three different lighting conditions of 70 Lux (Photopic), 20 mLux and 2 mLux (Scotopic). Based on parameters reported previously in similar behavioral experiments (*Umino et al., 2008*), bar stripe patterns were presented at a speed of 15 deg/s for various spatial frequencies ranging from 0.01 to 0.44 cycles/deg in photopic condition. For the scotopic conditions, the stimulus was maintained at a constant temporal frequency of 0.73 Hz and spatial frequency in the range 0.01–0.3 cycles/deg. The temporal frequency here refers to the combination of spatial frequency (cyc/deg) and speed of movement in (deg/s), which gives an effective temporal frequency, namely the change of contrast at a given point on the screen, which was maintained constant at a particular temporal frequency (0.73 cyc/s or Hz). The contrast of the object displayed on the screen was in the range 100–2%. For determining the threshold contrast, display contrast was reduced in steps of 5% up to absolute contrasts of 10% and steps of 2% below 10% contrast. The sizes of stripes tested were (6, 8, 11, 22, 33, 44, 55, 66, 88, 95, 100, 106 cycles/360°). Each stimulus was presented for a total of 30–35 s in sets of 5 s each with a gap of 5 s between each presentation. The direction of rotation of the stripes was altered between left and right for each subsequent trial and chosen at random for trial-1. Once the threshold contrast was experimentally determined, for statistical analysis purpose, response values for contrasts above the threshold were designated to be 'yes' response and values below the threshold as 'no' response.

## Nocturnal adaptation of behavioral testing setup

The ambient lighting of the test chamber for photopic condition was measured using a Lux meter (Testo 540). To reduce the lighting to scotopic levels appropriate ND filter sheets (ND 3.6, ND 4.8) were placed on the monitors. The ND filters were assembled by combining ND-1.2 filter sheets (e-color+ #299, Rosco Laboratories Inc). A custom made infrared light source was also installed to monitor and enable tracking of the animals on the platform under scotopic conditions. The Rho-/- mice were used as a control to ensure that the responses of the mice in the scotopic display test conditions purely relied on the rod visual pathway.

## Modelling and simulation of light propagation

### 2-photon mapping of ONL model

Wild-type C57BL/6J mice were sacrificed by cervical dislocation. Immediately, eyes were enucleated and then cut in half around the equator, discarding all components of the eye but the posterior eye-cup. Retina was peeled off from the eye-cup. The retinal isolation was performed in paraformalde-hyde (PFA) 4% in phosphate-buffered saline (PBS) solution and then left suspended to complete fixation for 20 min. The sample was then transferred to a PBS solution at 4°C after fixation. The fixed sample was deposited inside a TEFLON container and embedded in low melting agarose. The aga-rose embedded sample was sectioned adapting the method described previously (*Clérin et al., 2014*). The resulting retinal cross sections were stained with Hoechst 33342 and then wet mounted in a 50% glycerol/PBS solution using a No. 1 cover slip (Corning Inc, USA). Imaging was performed with confocal microscope (LSM 780, Zeiss Germany) in two-photon mode, equipped with a tunable pulsed infrared laser (Chameleon Vision II, Coherent, US) (excitation wavelength 730 nm, Objective: Zeiss LCI Plan-Neofluar 63x/1.3). The acquired intensity image was of size $190 \times 190 \times 82$ μm with pixel-sizes of $83 \times 83 \times 250$ nm in lateral dimensions and in depth.

### Image processing and segmentation of ONL model

To create a realistic refractive index map of packed nuclei within the ONL, the intensity image was first segmented into nuclei regions. To that end, a random forest classifier was trained via Fiji (*Arganda-Carreras et al., 2017*; *Schindelin et al., 2012*) to densely classify each pixel into back-ground or foreground (nuclei). A watershed segmentation (*van der Walt et al., 2014*) was then applied on the probability map with manually generated seed points, resulting in 1758 individual nuclei instances. The refractive indices for these phases have been carefully estimated previously in single cell studies (*Błaszczak et al., 2014*; *Kreysing et al., 2010*; *Schürmann et al., 2017*; *Solovei et al., 2009*). Finally, the refractive index distribution inside each nuclei region was gener-ated according to the two different models:

Inverted: Consisting of two refractive phases with n1 = 1.357 and n2 = 1.382, corresponding to euchromatin and heterochromatin, respectively. Each nuclei mask was split into shell and core regions of equal volume (via morphological shrinking operations on each mask), which were then assigned the respective refractive indices (n1 for shell, n2 for core).

Chromocenter: Here, 8–12 chromocenters were randomly picked within the nuclei mask and assigned points close to either the nuclei border or those chromocenters to the high refractive index phase (n2) until its joint volume reached half the full nuclei volume. The other points were then assigned the less dense refractive index n1.

The resulting refractive index distribution was then blurred in both cases with a small gaussian (sigma = 2 px) to create a smooth distribution. For both models it was furthermore ensured that both refractive phases occupied the same total volume.

### Light propagation simulations and scattering

Light propagation through both ONL models was simulated with GPU-accelerated scalar beam propagation method (*Weigert et al., 2018*). A computational simulation grid of size (1024,1024,645) with pixel-size 83 nm was used and the propagation of a plane wave (wavelength 500 nm) through the different ONL refractive index distributions was simulated. The surrounding was assumed to have a refractive index of $n_0 = 1.33$. The integrated side scattering cross sections were calculated from the angular spectrum as per previous reports (*Weigert et al., 2018*).

### Relative contributions to MTFs from ONL & outer segments

In order to assess relative contribution of ONL and outer segments to the MTF of the retina dedi-cated simulations were carried out. These compared the scattering from chromocenters in an ONL (1 or 8 per nucleus), with outer segments that were simulated as cylinders that were 1.6 μm in diam-eter and 25 μm in length. The refractive index of the core of the outer segments was assumed to be 1.42 (*Sidman, 1957*). For the recorded simulations the scattering anisotropy factor and efficiency were extracted and converted into a frequency domain MTF data using appropriate theoretical models (*Henyey and Greenstein, 1941*; *Wells, 1969*). Results show that outer segments only have a negligible impact on the overall MTFs (*Figure 3—figure supplement 2G*), in agreement with

previous experimental findings (*Enoch, 1963*) and models of the outer segment (*Vohnsen, 2007*; *Vohnsen, 2014*) acting as waveguides.

## Measurements of ocular parameters

Freshly excised mouse eye balls were imaged under a Olympus stereo microscope SZX16 equipped with a Q-imaging camera. The lens was also imaged under dark field conditions to better visualize the lens periphery. From the recorded images, the ocular parameters - lengths of the eye along two orthogonal axes were measured manually using FIJI. For the lens, mean feret diameter from the contour of the lens periphery was measured as an average estimate of the size of the lens.

## Data and materials availability

Data and specifications of simulations supporting the findings of this study are available via https:// dx.doi.org/10.17617/3.3a. The biobeam software is available publicly from: https://maweigert. github.io/biobeam.

## Acknowledgements

We would like to acknowledge the help from the following facilities at MPI-CBG - biomedical services, transgenic core, DNA sequencing, cell technology and FACS, light-microscopy during multiple phases of the project. The authors thank Caren Norden, Jochen Guck as well as Thomas Cremer, Jayaram K Iyer, Elisabeth Knust, Iain Patten, Andreas Reichenbach and Marino Zerial for helpful discussion and comments. The authors thank OptoDrum, Striatech GmbH for assistance in adapting the optomotor setup (OptoDrum) to scotopic light levels.

## Additional information

### Funding

| Funder | Grant reference number | Author |
|---|---|---|
| Max-Planck-Gesellschaft | | Kaushikaram Subramanian<br>Martin Weigert<br>Heike Petzold<br>Alfonso Garcia-Ulloa<br>Eugene W Myers<br>Moritz Kreysing |
| Technische Universität Dresden | | Oliver Borsch<br>Marius Ader |
| Deutsche Forschungsgemeinschaft | AD375/6-1 | Oliver Borsch<br>Marius Ader |
| Bundesministerium für Bildung und Forschung | 031L0044 | Kaushikaram Subramanian<br>Eugene W Myers<br>Moritz Kreysing |
| Deutsche Forschungsgemeinschaft | SO1054/3 | Irina Solovei |
| Deutsche Forschungsgemeinschaft | FZT111 | Oliver Borsch<br>Marius Ader |
| Deutsche Forschungsgemeinschaft | EXC68 | Oliver Borsch<br>Marius Ader |
| Deutsche Forschungsgemeinschaft | SFB1064 | Irina Solovei |
| European Research Council | 853619 | Kaushikaram Subramanian<br>Moritz Kreysing |

The funders had no role in study design, data collection and interpretation, or the decision to submit the work for publication.

## Author contributions
Kaushikaram Subramanian, Conceptualization, Data curation, Formal analysis, Validation, Investigation, Visualization, Methodology; Martin Weigert, Data curation, Software, Formal analysis, Validation, Investigation, Visualization, Methodology; Oliver Borsch, Software, Validation, Investigation, Methodology; Heike Petzold, Data curation, Validation, Investigation, Methodology; Alfonso Garcia-Ulloa, Investigation, Methodology; Eugene W Myers, Software, Supervision, Funding acquisition, Validation, Methodology, Project administration; Marius Ader, Conceptualization, Resources, Data curation, Supervision, Funding acquisition, Validation, Project administration; Irina Solovei, Conceptualization, Resources, Data curation, Supervision, Funding acquisition, Validation, Investigation, Visualization, Methodology, Project administration; Moritz Kreysing, Conceptualization, Resources, Data curation, Formal analysis, Supervision, Funding acquisition, Validation, Visualization, Methodology, Project administration

## Author ORCIDs
Kaushikaram Subramanian (iD) https://orcid.org/0000-0002-9050-5672
Marius Ader (iD) http://orcid.org/0000-0001-9467-7677
Moritz Kreysing (iD) https://orcid.org/0000-0001-7432-3871

## Ethics
Animal experimentation: All animal studies were performed in accordance with European and German animal welfare legislation (Tierschutzgesetz), the ARVO Statement for the Use of Animals in Ophthalmic and Vision Research, and the NIH Guide for the care and use of laboratory work in strict pathogen-free conditions in the animal facilities of the Max Planck Institute of Molecular Cell Biology and Genetics, Dresden, Germany and the Center for Regenerative Therapies Dresden, Germany. Protocols were approved by the Institutional Animal Welfare Officer (Tierschutzbeauftragter) and the ethics committee of the TU Dresden. Necessary licenses 24-9168.24-9/2012-1, DD24.1-5131/451/8 and TVV 16/2018 (DD24-5131/354/19) were obtained from the regional Ethical Commission for Animal Experimentation of Dresden, Germany (Tierversuchskommission, Landesdirektion Sachsen).

## Decision letter and Author response
Decision letter https://doi.org/10.7554/eLife.49542.sa1
Author response https://doi.org/10.7554/eLife.49542.sa2

# Additional files

## Supplementary files
• Transparent reporting form

## Data availability
Data and specifications of simulations supporting the findings of this study are available via: https://dx.doi.org/10.17617/3.3a. The biobeam software is available publicly from: https://maweigert.github.io/biobeam.

The following dataset was generated:

| Author(s) | Year | Dataset title | Dataset URL | Database and Identifier |
|---|---|---|---|---|
| Kaushikaram Subramanian, Martin Weigert, Oliver Borsch, Heike Petzold, Alfonso Garcia-Ulloa, Eugene W Myers, Marius Ader, Irina Solovei, Moritz Kreysing | 2020 | Rod nuclear architecture determines contrast transmission of the retina and behavioral sensitivity in mice | https://dx.doi.org/10.17617/3.3a | Edmond, 10.17617/3.3a |

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
