## [Decision Letter]

Thank you for submitting your article "Rod nuclear architecture determines contrast transmission of the retina and behavioral sensitivity in mice" for consideration by *eLife*. Your article has been reviewed by three peer reviewers, and the evaluation has been overseen by a Reviewing Editor and Ronald Calabrese as the Senior Editor. The following individuals involved in review of your submission have agreed to reveal their identity: Austin Roorda (Reviewer #1).

The reviewers have discussed the reviews with one another and the Reviewing Editor has drafted this decision to help you prepare a revised submission.

As you will see, all of the reviewers were impressed with the significance and thoroughness of your work. All three reviewers also had specific and useful comments for improving the manuscript. Among them are three general suggestions to which we would like to draw your attention:

1) The Materials and methods section needs substantial editing for clarity and detailed methods of how forward/side scatter measures were performed. A large portion of the manuscript depends on this analysis and it is imperative to include the details.

2) Include a rigorous description of which tests are more or less behaviorally relevant for mouse vision. Showing data for ~1 cycle/deg may exaggerate the biological benefit, as does modelling a ~3 micron point spread function at the retina. The cat face may be marginally relevant based on visual angle on the retina, although it has a certain charm for the non-expert reader.

3) We encourage you to discuss other possible interpretations of the data, including competing hypotheses for the role of nuclear inversion.

Reviewer #1:

The authors have expanded on their previous work to understand the functional significance of the `inversion' of the nuclear architecture in nocturnal mammals, specifically, in this case, the mouse. The conclusion is that the inverted nuclear structure minimizes side scattering, and facilitates forward scattering with a resulting benefit that higher contrast images can reach the photoreceptors, thereby improving contrast sensitivity.

While readers might have inferred this conclusion based on earlier papers by members of this team, the authors do a very nice job of confirming it by comparing contrast transmission and behavioral performance between wild-type mice and mice with a genetic modification (TG-LBR) that prevents the 'inversion' from taking place (these TG-LBR mice appear to be otherwise unaffected visually). The wild-type mice show better contrast sensitivity of a magnitude of 18% and 27% for scotopic (nighttime) light levels compared to the TG-LBR mice. Interestingly, the wild-type and TG-LBR mice behave similarly under photopic (daylight) conditions, which the authors sensibly attribute to a reduction in noise caused by high-photon flux.

The improvement in performance of 18-27% is modest, but not negligible. The authors show that these modest improvements in contrast sensitivity serve to increase detection probabilities many-fold at dim, near-threshold levels. Therefore, the functional advantages of the nuclear inversion are convincing.

The Materials and methods section is very sloppy and needs to be revised. Also, there are some details missing (eg how forward and side-scatter is measured). Otherwise, the paper is well-written, the science is solid, and it sheds new light on the fascinating process of retinal development.

1) Abstract: `…retinal optical quality improves 2-fold…'. The authors overstate the optical benefit by choosing to report on one metric, which was the ratio of the areas of the MTF between the wild type and TG-LBR mice. This is an odd choice, because most of the spatial frequencies used for this metric are seemingly irrelevant for mouse vision. It would be more appropriate for the authors to provide in the abstract numbers for the behavioral improvements (18-27%)

2) Abstract: there should be no hyphen in `contrast-transmission'. (here and throughout the document)

3) Introduction paragraph one: what does less-dense mean? Are the authors referring to refractive index, optical density or actual density?

4) Results paragraph two and three and subsection “Flow cytometry”: Since it is so critical for this paper, it would be helpful if the authors could briefly describe how forward- and side-scattering are measured rather than just providing a citation.

5) Results paragraph three: The definition of side-scatter is vague. Here the authors define it as narrow scattering at 90 deg, but later (eg in subsection “Improved retinal contrast transmission”) they define it as scattering at angles > 30 degrees. Also the authors need to define the axis labels `Forward Scattering Area' and `Side-Scattering Area' in figure 1.

6) Figure 1C (inset). What does Volume-specific scattering mean? This needs to be defined.

7) Figure 1G: What do the rectangles in Figure 1G represent? Are they just sketched in or do the dimensions have an important meaning.

8) Subsection “Improved retinal contrast transmission” –, Figure S5: The authors state that they mimic the mouse eye by using an optical system with a similar f-number. But in the next paragraph, they state that the MTFs '…do not display a strict resolution limit.' These are conflicting statements. The use of limited aperture in the system means that it will have its own MTF. The authors should show the optical system MTF in their plots on Figure 3.

9) In the same subsection: The initials T.V. should be deleted.

10) What range of spatial frequencies were used for these computations?

11) Subsection “Improved retinal contrast transmission”, Figure 3:D2 and D3, subsection “PSF measurements”: The intensity of the PSF in the figure is lower for the TG-LBR mouse across the entire displayed range of -20 to 20 microns. But the authors state that the integrated intensity is the same between the two when the PSF is integrated over an 80 x 80 micron area. I am very skeptical that the integrated intensity under the two curves in Figure 3:D2 will become equal.

12) Results section final paragraph: "This suggest…."

13) Discussion paragraph one: The lack of `nuclear inversion' in diurnal animals is intriguing and the authors make a very sensible suggestion that the ONL is significantly thinner in diurnal animals. However, that statement should be backed up by proper citations or, better yet, a table or a plot comparing ONL between nocturnal and diurnal animals.

14) Materials and methods: In general, this section is sloppily written with numerous typos, combinations of present and past tense – often in the same sentence – and unclear writing. There are numerous typos. The authors flip between the abbreviation SR and strehl ratio.

15) Subsection “Calculation of MTF”. How do the authors propose to use this technique to measure optical impact of outer segments? Note that ex vivo preparations are vulnerable to optical artifacts, especially the delicate optical properties of the retina.

16) Behavioral assessment: What does the temporal frequency mean? Was the stimulus flickering? Or moving, or both? This entire section is very poorly written.

17) Subsection “Image processing and segmentation of ONL model”: Why was this smoothing necessary? Were the final results different when they were not smoothed? Does the smoothing generate refractive index profiles that are more realistic?

18) Subsection “Relative contributions to MTFs from ONL and outer segments”: Replace OS with 'outer segment'

Perhaps the Matlab script mentioned in the text should be shared.

Reviewer #2:

Paper Summary:

The authors build on a body of literature that has identified the interesting phenomenon of "nuclear inversion" in nocturnal mammals. In this report, the authors test the hypothesis that the re-organization of euchromatin and heterochromatin within the nucleus of rod photoreceptor cells could serve to benefit nocturnal mammals by reducing scatter in the outer nuclear layer which is thick in rod-dominant mammals such as mice. An impressive set of data is collected in the report. The authors interpret their findings as supportive of a role of improved contrast sensitivity due to nuclear inversion which purportedly reduces optical scatter, and thereby improves the contrast ratio of images that must project through all retinal layers before striking the outer segments of rods.

The paper is thoughtfully composed and was generally a pleasure to read. The data set is impressive and authors are congratulated on a wholesome battery of tests that span in vitro preparation, phantom simulations, mouse behavioral testing, histology with immunolabeling and transgenic animals that support the general hypothesis. The major criticism for the report, however questions the very raison d'etre of the manuscript; "just how beneficial is this nuclear inversion to mouse visual performance?" While nuclear inversion is indeed a strange behavior of outer retinal cells (especially rods), it is unclear whether this is an epiphenomenon of some other function important to rods, or whether, as the authors would suggest, truly provides visual contrast benefit to the animal. The authors provide some evidence in support of this idea, but there are several misleading conclusions drawn from figures (especially Figure 3) which overstate the contrast benefit to mice by using simulations that are not behaviorally relevant.

Problem 1: Authors show the MTF improvement of contrast transmission when projecting sinusoidal patterns directly onto the retina. The differences in retinal contrast appear impressive in Figure 3AB. When comparing pups or TG-LBR mice (which also do not have nuclear inversion) to the adult WT mice that do have nuclear inversion, contrast transmission appears to increase. However the range of spatial frequencies tested are not generally thought to be behaviorally relevant to mice. Reports by Histed MH, Carvalho LA, Maunsell JH. (J Neurophysiol 2012, and corroborated by a multitude of other studies) suggest that maximum spatial frequency cutoff for the mouse is near 0.5 cyc/deg. This represents the very lowest of the tested spectrum in Figure 3AB. By those measures, roughly 2/3 of the data is behaviorally irrelevant to the normal mouse. When considering data from 0-0.5 cycles/degree, the effect is visually modest in comparison. Reviewer requests revision of the figure to reflect the improvement range to that closer of what is relevant to mouse visual behavior.

Problem 2: Projection of 3 micrometer PSF into the mouse retina (Figure 3D) is behaviorally irrelevant. Based on the literature that the authors cite (and more), Geng et al., Schmucker and Schaeffel, 2004 and others such as Remtulla and Hallett, 1985, a 3 micron PSF is a highly unnatural stimulus for the mouse retina because of spherical aberration, longitudinal chromatic aberration, transverse chromatic aberration, a constantly growing mouse eye and an optically thick retina. Anything less than a single-wavelength stimulus therefore would be impossible to naturally project at a 3 micrometer spot, and thus it is unclear why the authors are using this highly unnatural stimulus to model the PSF spread in Figure 3D.

Problem 3: Authors attempt to simulate the behavioral benefit to the mouse by a friendly example of what the mouse would "see" in an approaching cat by showing a phantom of the cat face. This is a fun example, but again represents a scenario that is unlikely due to the visual acuity of the mouse (adult or otherwise). If assumed that behavioral spatial frequency is limited to ~0.5 cyc/deg, there is little chance the mouse would visualize the cat eyes at any distance represented by Figure 3. The reviewer calculates that interpupillary distance of a typical house cat (which is assumed to be a biotypical natural predator of the mouse? certainly not a tiger!) is 36 mm (following Hughes, 1972 Vision Research). If we are generous and round this to 4cm, the subtended angle on the mouse retina will surely not render the eyes of the cat in such a way that the authors illustrate. At 4 meters, subtended angle is nearly 0.57degrees. At 2 meters, subtended angle is 1.14 degrees. Again, this far exceeds the reported visual acuity of the mouse and therefore the example is inappropriate, behaviorally irrelevant and is misleading to the general scientific audience. There would be no visual benefit to the mouse in these conditions even if nuclear inversion were found to benefit contrast transmission. Request removal of this figure.

Problem 4: Problems 1-3 are further compounded that the generous spatial frequency cutoff for the mouse is 0.5 cycles/deg for photopic conditions (Prusky et al., 2000; Histed et al., 2012). Spatial frequency tuning for the WT mouse is considerably worse under scotopic conditions which is the regime that stands to benefit from rod nuclear inversion (authors report this is a rod-dominated effect and cones generally do not show such behavior). Umino, Solessio and Barlow, 2008, show scotopic contrast sensitivity is even lower than photopic in the mouse. Behaviorally tested cutoff is near 0.2 cyc/. When this is projected back on to the data from Figure 3AB,D1,D2,E and F) the behavioral benefit in Figure 3 seem to be baseless.

Despite these shortcomings, the manuscript has merit. Problems 1-4 are somewhat mitigated by compelling data in Figure 4 which do show a slight benefit in WT mice (with nuclear inversion) vs LBR mice which presumably do not. Scientific audience is left to trust that TG-LBR mice have otherwise normal ocular behavior with the exception of high-chromocenter rod nuclei. Further description of the phenotype would convince skeptics further (including eye size and anterior optical media clarity which could also account for the result in Figure 4).

In the discussion, the authors do not provide enough latitude that other epiphenomenon and bioselection-driven reasons for nuclear inversion are possible. The manuscript would be stronger if such openings for these possibilities are explored further. The reader is left with the feeling that the problem is solved, which it is not. Data is provided to support a hypothesis.

Figure 4F not described in Figure 4 caption.

Supplementary data is appropriate and impressive.

Reviewer #3:

The work by Subramanian et al. demonstrates that the inversion of nuclear architecture in the rod photoreceptors of mice improves visual function in dim light conditions. The paper is very well-written and easy to follow. The work is of the highest quality and the well-thought-out experiments nicely support the conclusions. It was a fun paper to read!

The strength of the paper comes from using a range of approaches, whole animals, tissue histology, dissociated cells, excised retinas, in vitro model systems, and theoretical calculations to demonstrate not only that the inversion improves visual function, but also provide a clear mechanistic explanation. Very convincing!

---

## [Author Response]

As you will see, all of the reviewers were impressed with the significance and thoroughness of your work. All three reviewers also had specific and useful comments for improving the manuscript. Among them are three general suggestions to which we would like to draw your attention:1) The Materials and methods section needs substantial editing for clarity and detailed methods of how forward/side scatter measures were performed. A large portion of the manuscript depends on this analysis and it is imperative to include the details.

The Materials and methods section has been updated to provide clear and detailed descriptions, and to address the reviewers’ comments. A summary of the changes in the Materials and methods contained in the revised manuscript is provided below.

1) The description of the FACS experiments has been expanded as follows:

“Flow cytometry

FACS scattering analysis of retinal cells and sorts according to light scattering profiles were performed based on previously published methodologies (Feodorova et al.,

2015). […] Volume-specific scattering thus refers to the light scattering normalized by the amount of material, used to compare the light scattering by a material of given volume/mass but different size distribution.”

2) The Matlab scripts used for the scattering calculations and the parameters used have been added.

“Mie models of nuclei

The scattering intensity calculations for the multichromocenter-nuclei depicted in Figure. 1H, were performed using Mie scattering models of spheres in a refractive index contrast of 2% (Kreysing et al., 2010), the reported contrast of refraction between heterochromatin and euchromatin. […] Relative scattering efficiencies for packed scatterers represented in Figure 3—figure supplement 2E were calculated based on dependent scattering models (Twersky, 1978).”

3) The grammatical and typographic errors have been corrected.

4) The description of the behavioural studies has been revised and extended to include relevant details and revised for clarity:

“Behavioral assessment – Optomotor response

Visual behavioral response was assessed using a fully automated, monitor based optomotor drum setup obtained from Striatech (Striatech GmbH, Tübingen, Germany) and the experiments were conducted at the CRTD, TU Dresden, Germany.[…] Once the threshold contrast was experimentally determined, for statistical analysis purpose, response values for contrasts above the threshold were designated to be “yes” response and values below the threshold as “no” response.”

5) Based on requests by the reviewers to investigate differences in the ocular parameters between WT and TG-LBR mice, the Materials and methods section also now contains a description of these experiments:

“Measurements of ocular parameters

Freshly excised mouse eye balls were imaged under a Olympus stereo microscope SZX16 equipped with a Q-imaging camera. The lens was also imaged under dark field conditions to better visualize the lens periphery. From the recorded images, the ocular parameters – lengths of the eye along two orthogonal axes were measured manually using FIJI. For the lens, mean feret diameter from the contour of the lens periphery was measured as an average estimate of the size of the lens.”

2) Include a rigorous description of which tests are more or less behaviorally relevant for mouse vision. Showing data for ~1 cycle/deg may exaggerate the biological benefit, as does modeling a ~3 micron point spread function at the retina. The cat face may be marginally relevant based on visual angle on the retina, although it has a certain charm for the non-expert reader.

Thank you for raising these issues. We fully agree that the assessment of the optical properties of the retina partially exceeds the visual acuity of mice, particularly the ability to resolve moving stripe patterns under low light conditions. As we explain below in some detail, however, we think there is good reason to show some of the full data sets (i.e. the MTF curves). Nevertheless, we also agree that we could have done a better job of explaining their relevance. Also, other optical data was insufficiently motivated in the text (e.g. the point sources), and in one case required an adjustment in measurement conditions (the cat approach).

There are a variety of visual behavioural tests to assess the performance of the visual system (Pinto and Enroth- Cugell, 2014). Of these, the use of the optomotor response has been shown to best determine the differences in visual capabilities of mice taking into account the complete process of visual perception in the brain (Gasparini et al., 2019; Huberman and Niell, 2011). The similarity between the stimulus and pattern of images to assess the retinal optical quality and the behavioural sensitivity thus enables direct determination of the role of retinal optics in visual perception. The representation of the results and the rationale behind the optical experiments) have now been appropriately explained in the revised manuscript (see below for dedicated explanations on MTF, PSF and cat images).

**Complete MTF representation:**

We fully agree with the view that mice do not show behavioural responses to stripe patterns (especially sinusoidal ones) with frequencies above 0.5 cycle/deg, particularly not under low light conditions. We also agree that our analysis of the optical quality of the retinal tissue included contrast transmission at much higher spatial frequencies, the reasons for which are clarified later in this section. To start with, we included an additional panel focussed on the retinal image transmission capabilities in the visually relevant range for mouse (Panel C). To the same end, we recognised that a logarithmic scale is not very well suited to access the magnitude of the differential contrast transmission at spatial frequencies that were used during behavioural experiments. To make the retinal transmission data more accessible within this spatial frequency regime, we have now included a figure panel for low spatial frequency (coarser stripes) data in a linear scale (manuscript Figure 3C). We believe this will help readers to better understand which region of the MTF curves overlap with behaviour behavioural response in mice.

From this, it can readily be seen that the contrast transmission is about 33% greater in WT compared to TG-LBR at frequencies where the WT mouse has the greatest advantage over the TG-LBR mouse in our behavioural tests. Under these conditions, there is also a 45% greater contrast transmission than observed in a WT pup. The revised manuscript now has additional text describing the contrast loss in images in the regime relevant for mouse behavioural tests:

“Notably, when we focus on the retinal transmission data within the spatial frequency regime that is relevant for mouse vision (Alam et al., 2015; Prusky et al., 2004; 2000), (Figure 3C) it can readily be seen that the contrast transmission is up to 33% greater in WT compared to TG-LBR at frequencies ~0.28 cycles/deg. Equally, the contrast transmission in this behaviourally relevant regime also increases up to 45% in adult compared to the pups.”

We hope this modification allows a better comparison of the MTF curves with the behavioural data and provide a convincing argument of the causal relationship between them. Nevertheless, we would also like to explain our rationale to show the complete data rather than just a subset. Owing to the broader interest of an interdisciplinary readers and the broader implications of the retinal tissue as an “optical model tissue”, the complete presentation of the findings seems justified to us. In particular:

a) An MTF curve provides information on the mechanism of the contrast reduction. In particular, our data shows that there is no frequency cut off, which speaks for a contrast reduction due to image veil caused by high-angle scattering. We think that showing the full data here will help readers to interpret the scattering characteristics of the rod nuclei and the outer nuclear layer described in the sections of the manuscript preceding the results on retinal image transmission.

b) The long tail of transmitted frequencies that we show further explains why, despite a rapid reduction of contrast with increasing frequencies, the photoreceptor mosaic can be reconstructed in an OCT measurement. Although not explained in our paper in great detail, we think that the display of the full MTF data addresses two very different aspects on the topic of retinal optics: (i) differential transmission at low and mid-range frequencies due to nuclear inversion, (ii) the ability to reconstruct rod mosaic images at much higher resolution than relevant for vision (depending on confocal detection of scattered photons at very high frequencies). OCT imaging is clearly not the topic of this paper, but in our experience, ophthalmologists welcome this side reference.

c) Furthermore, MTFs and Strehl ratios are some of the canonical metrics to quantify an optical system. Especially the quantification of Strehl ratios is one of the most common and comprehensive measures of an optical system performance. Thus, we think from an optical point of view it is imperative to include the measurable contrast transfer information from all the contributing spatial frequencies.

**PSF measurements:**

Regarding the transmission of a 3μm PSF, we fully agree that the improved transmission cannot be completely related to any functional consequences, and we clarify that such an inference was not intended in the manuscript. Furthermore, we agree with the reviewers that the mouse lens is not likely to be of sufficient quality to project such a sharp image that the differential peak intensities, which we read out as the Strehl ratio, would be of physiological relevance. At the same time, we feel, it is still worthwhile presenting this data because:

i) increased the peak intensity are predicted by the volume under the MTF curve, and we think it showing this consistency further increases some readers confidence in our data.

ii) As stated in the manuscript, this data is suitable to show that the resolution of retinal transmission is not strongly affected, thereby ruling out a different and alternative optical explanation for the occurrence of nuclear inversion.

iii) Lastly, knowing the aberrations of the lens, the additional explicit knowledge of the PSF of the retina might be very useful for improved theoretical predictions of image quality at the level of photoreceptor cells. In other words, it is still widely thought that the retina is completely transparent, and that image quality is mostly determined by the lens. We think showing the PSF of the retina alone will also help to refine this over-simplified picture.

To make sure there are no misinterpretations, we modified the relevant passage of the manuscript accordingly by adding:

“The measured resolution based on full-width half maximum (FWHM) of the PSF, however, remained virtually unchanged did not show any differences (4.32 ± 2.38 𝜇m, 3.75 ± 2.01 𝜇m, mean ± SD for WT and TG-LBR retinae respectively, Figure 3 E2), especially no changes that could physiologically impact acuity, which is known to be significantly lower in mouse. In addition to independently corroborating our MTF measurements, these results emphasize that nuclear inversion enhances contrast transmission through the retina but is unlikely to benefit acuity. From a mechanistic point of view, these measurements indicate that contrast is lost due to the generation of image veil from side scattering, which overcasts attenuated, but otherwise unchanged signals.”

To conclude, we would like to reiterate that we are aware that the increase in retinal optical quality is substantially greater than the integrated sensitivity (albeit not greater that the maximum sensitivity increments at critical conditions). However, for the reasons listed above, we would like to retain the physical description of the tissue as an optical medium as presented.

**Cat image:**

We thank the reviewers for bringing up the significance of this data. We included the data to bridge the gap between the artificial sinusoidal stripe patterns and real-life images. The motivation was simply to illustrate that also real-life images are transmitted at increased contrast. In search of an image for this, we thought a cat might be appropriately illustrative and we generally received very positive feedback when showing this to colleagues in the field. However, the reviewers are completely correct that the choice of distances was not very well backed up by existing knowledge about mouse visual capabilities, especially acuity cut-off under low light conditions.

We therefore repeated the analysis at frequencies below 0.5 cycles/degree (namely 0.1, 0.2 and 0.3 cycles/ degree, revised manuscript Figure 3G). According to the literature provided by the reviewers, this shifts the assay into a behaviourally relevant regime. New results show a comparable advantage at distance below 1 meter. The only difference in the new measurements is that intensities below distance of approximately 0.5m saturate for WT mice, which however does not take away from the original idea of improved predator images at the level of the photoreceptors. Although our study does not address whether a mouse first sees or smells or hears a cat (or about the abundance of domesticated cats during mouse eye evolution), it does fulfil the purpose of illustrating the effects with a real-life image. The experiments with the approach of a predator by projection of a cat face have now been updated for the visually relevant distances. The section describing these results have also been modified for clarity and relevance of these measurements:

“An advantage of improved retinal contrast transmission is suggested when following the motion of individual (nonaveraged) light stimuli that appear at considerably higher signal-to-noise ratios (Figure. 3F) at the outer segments level. A putative visual advantage to appropriately scaled real-life examples, such as images of an approaching cat micro-projected through a mouse retina is illustrated in Figure 3G. Nuclear inversion results in cat images becoming visible considerably earlier compared to mice that lack nuclear inversion (0.70 vs 0.45 meters, at a given arbitrary noise threshold). These results suggest that nuclear inversion may offer enhanced visual competence that originates from improved contrast preservation in retinal images. More objective and established methodologies to test the impact of nuclear inversion for actual behaviour is addressed in the next section.”

3) We encourage you to discuss other possible interpretations of the data, including competing hypotheses for the role of nuclear inversion.

The reviewers are right, of course, that we cannot exclude other functions of nuclear inversion. Mammalian, eyes are amongst the most complex organs we know, and definitely the retina and nuclear architecture therein are multi-constrained systems. This raises various relevant questions:

i) Does the improved contrast sensitivity stem from optical change downstream of nuclear inversion, or could there be other reason for the differential sensitivity?

ii) Why aren’t photoreceptor nuclei of diurnal animals inverted?

iii) Despite the direct link between enhanced optical properties and improved sensitivity, does nuclear inversion also have other functions?

We have touched upon question (i) already in the existing manuscript with a rescue experiment in which we demonstrated sufficiency of increased retina contrast transmission to explain improved contrast sensitivity. We would like to apologise that the relevant figure cation that described the results of the rescue experiments were inadvertently cropped in the manuscript that we submitted for review. We think that this missing caption might have prevented reviewers from engaging with these rescue experiments. Furthermore, we have performed additional control experiments as suggested by the reviewers and establish that the ocular anatomy of the WT and TG-LBR mice are not significantly different and do not contribute to any possible changes in the behavioural sensitivities. We have provided this data in supplementary figures. We now include the 3 missing lines of figure caption (manuscript Figure 4F), and we also expanded the paragraph explaining the rationale behind these rescue experiments (underlined explanations were added):

“Finally, we asked whether reduced visual sensitivity of mice lacking the inverted nuclear architecture can be sufficiently explained by inferior contrast transmission of the retina. […] Thus, improved retinal contrast transmission is indeed sufficient to explain increased contrast sensitivity in mice.”

Question (ii) had already been addressed by explaining that there would be reduced benefit, for which we provide a detailed discussion and illustration in Figure 4—figure supplement 1. Some additional arguments have been presented in this regard in the revised manuscript:

“Besides a reduced need for inverted photoceptor nuclei in diurnal mammals, reduced efficiency of canonical DNA repair mechanisms (Frohns et al., 2014) in highly condensed chromocenters, could mean a significant disadvantage in the diurnal retina and susceptibility to stress and degeneration (Boudard et al., 2011; Dyer, 2016), could also mean a significantly higher cost for inverted nuclei in diurnal species, as their retinae are intrinsically strongly exposed to high-energy, ultra-violet photons. In conclusion, we showed that rod nuclear inversion is necessary and sufficient to explain optically enhanced contrast sensitivity in mice. Our work thereby adds functional significance to a prominent exception of nuclear organization and establishes retinal contrast transmission as a new determinant of mammalian fitness.”

Question (iii) is a very valid concern that we have now addressed. For this, we have expanded the Discussion with the following additional paragraph to discuss other interpretations and competing hypotheses.

“As mammalian eyes are evolutionarily multi-constrained systems, one could ask if nuclear inversion might also serve other functions beyond the improved contrast sensitivity that we have showed. […] While these additional functions of nuclear inversion currently remain speculations, it is worth reflecting about the relevance of the visual benefits demonstrated here for enhancing animal vision in general.”

Reviewer #1:The authors have expanded on their previous work to understand the functional significance of the `inversion' of the nuclear architecture in nocturnal mammals, specifically, in this case, the mouse. The conclusion is that the inverted nuclear structure minimizes side scattering, and facilitates forward scattering with a resulting benefit that higher contrast images can reach the photoreceptors, thereby improving contrast sensitivity.While readers might have inferred this conclusion based on earlier papers by members of this team, the authors do a very nice job of confirming it by comparing contrast transmission and behavioral performance between wild-type mice and mice with a genetic modification (TG-LBR) that prevents the 'inversion' from taking place (these TG-LBR mice appear to be otherwise unaffected visually). The wild-type mice show better contrast sensitivity of a magnitude of 18% and 27% for scotopic (nighttime) light levels compared to the TG-LBR mice. Interestingly, the wild-type and TG-LBR mice behave similarly under photopic (daylight) conditions, which the authors sensibly attribute to a reduction in noise caused by high-photon flux.The improvement in performance of 18-27% is modest, but not negligible. The authors show that these modest improvements in contrast sensitivity serve to increase detection probabilities many-fold at dim, near-threshold levels. Therefore, the functional advantages of the nuclear inversion are convincing.The Materials and methods section is very sloppy and needs to be revised. Also, there are some details missing (eg how forward and side-scatter is measured). Otherwise, the paper is well-written, the science is solid, and it sheds new light on the fascinating process of retinal development.

We apologise for numerous errors and partially lacking description of the methods. The Materials and methods section has now been updated to provide clear and detailed descriptions, and to address the reviewers’ concerns. A summary of the changes in the Materials and methods section is provided in response to the editors point 1.

1) Abstract: `…retinal optical quality improves 2-fold…'. The authors overstate the optical benefit by choosing to report on one metric, which was the ratio of the areas of the MTF between the wild type and TG-LBR mice. This is an odd choice, because most of the spatial frequencies used for this metric are seemingly irrelevant for mouse vision. It would be more appropriate for the authors to provide in the abstract numbers for the behavioral improvements (18-27%)

We agree that the chosen metric of optical improvements is not straight forward connected to the reported improvement of contrast sensitivity. However, we think that these findings on the overall improvement in the optical quality of retinal tissue are important and meaningful to the audience interested in tissue optics and microscopy. The similarity of the methods and test patterns used in the assessment of the tissue transmission and behaviour may indeed help draw correspondences between the improvements based on contrast and measures of image detail. The more general question of stimulus detectability (unlike resolvability) may take into account the retina’s ability to transmit higher spatial frequencies. Moreover, 18-27% integrated sensitivity improvement equally refers to one among many possible metrics; the response rates, for instance, in visually challenging regimes is shown to be enhanced by up to 10x (manuscript Figure 4E). Keeping these points in mind, we feel that reporting the numbers related to the independent observation of an optical quality improvement in a living heterogeneous tissue such as the retina, based on the most comprehensive metric (Strehl ratio), seems justified in the Abstract. To avoid any misleading implications for mouse vision, the numbers for behavioural improvements have now been added to the revised Abstract:

“Rod photoreceptors of nocturnal mammals display a striking inversion of nuclear architecture, which has been proposed as an evolutionary adaptation to dark environments. […] Our findings therefore add functional significance to a prominent exception of nuclear organization and establish retinal contrast transmission as a decisive determinant of mammalian visual perception.”

2) Abstract: there should be no hyphen in `contrast-transmission'. (here and throughout the document)

Thank you. This has been corrected in the Abstract. Corrections have been appropriately included in the revised manuscript.

3) Introduction paragraph one: what does less-dense mean? Are the authors referring to refractive index, optical density or actual density?

This is a good point. The density here refers to the actual mass density of chromatin, that are known also for other cell types (Imai et al., 2017). It is these increased mass densities that cause the RI differences earlier reported by the lead authors. The corresponding sentence in the revised manuscript has been modified to avoid any confusion.

“Interestingly, rod nuclei are inverted in nocturnal mammals (Błaszczak et al., 2014; Kreysing et al., 2010; Solovei et al., 2009; 2013), such that heterochromatin is detached from the nuclear envelope and found in the nuclear center, whereas euchromatin that has lower mass density (Imai et al., 2017) is re-located to the nuclear

periphery.”

Triggered by the reviewer’s remark, we noticed that we forgot to point at a related finding, which is that the heterochromatin core is of such high density that it even excludes free GFP molecules (Figure 1—figure supplement 1) The relevant description is now included in the Results section:

“This suggests that retinal cells are indeed optically specialized, as they scatter less light for a given size. This unique property for the rod cells could stem from the unusually dense packing of the heterochromatin in the centre of their nuclei, which notably even excludes free GFP molecules (Figure1—figure supplement 1B).”

4) Results paragraph two and three and subsection “Flow cytometry”: Since it is so critical for this paper, it would be helpful if the authors could briefly describe how forward- and side-scattering are measured rather than just providing a citation.

The details of the FACS measurements have been elaborated as indicated below and an extended description of the protocol has been added to the Materials and methods. Kindly also refer to our response to editors point 1.

“We then asked whether retinal cell somata are optically specialized with distinct light-scattering properties. […] Using forward scattering as a measure of cell size indicates that side scattering normalized by volume (volume-specific light scattering) is also noticeably lower in retinal cells (Figure 1C, inset).”

and

“In stark contrast however, sideward scattering, with a strong potential to diminish image contrast, was significantly reduced in adult retinal nuclei compared to the intermediate developmental stage (Figure 1F). Quantitative analysis of sorted nuclei from P25 retinae further revealed a monotonic relationship between chromocenter number and sideward scattering signal (Figure 1G). In particular, those nuclei with the lowest number of chromocenters were found to scatter the least amount of light.”

5) Results paragraph three: The definition of side-scatter is vague. Here the authors define it as narrow scattering at 90 deg, but later (eg in subsection “Improved retinal contrast transmission”) they define it as scattering at angles > 30 degrees. Also the authors need to define the axis labels `Forward Scattering Area' and `Side-Scattering Area' in Figure 1.

Thank you for pointing out the ambiguity. The terms forward and sideward scattering and the angles associated with their measurements are presented as defined in the commercial FACS system. Their definitions are now included in the text. Please refer to the response to reviewer1 comment 4 above. To avoid confusion, the side scattering term has been now changed to “large-angle scattering” in the section describing the results from simulations. We used simulations for extrapolation of angular light scattering that could not be captured in FACS measurements.

“These simulations suggest that especially the large-angle scattering (cumulative scattering signal at angles >30 deg) monotonically decreases when 10 chromocenters successfully fuse into one (Figure 2D2, 2E).”

6) Figure 1C (inset). What does Volume-specific scattering mean? This needs to be defined.

Volume-specific light scattering is the amount of light scattering per unit volume. The definition of the term has now been provided in the supplementary text.

“For the calculation of the volume specific scattering, the side scattering area was normalized by volume of nuclei by taking the forward scattering area as a measure for size. Volume-specific scattering thus refers to the light scattering normalized by the amount of material, used to compare the light scattering by a material of given volume/mass but different size distribution.”

7) Figure 1G: What do the rectangles in Figure 1G represent? Are they just sketched in or do the dimensions have an important meaning.

The description of Figure 1G is modified accordingly to include the following explanation.

“The rectangles represent sorting gates for microscopy analysis.”

8) Subsection “Improved retinal contrast transmission”, Figure S5: The authors state that they mimic the mouse eye by using an optical system with a similar f-number. But in the next paragraph, they state that the MTFs '…do not display a strict resolution limit.' These are conflicting statements.

We are grateful to the reviewer for pointing this out. As we clarify in the next point the MTF of the measurement system is there, and it has been taken in to account. Noteworthy however, compared to relevant frequencies transmitted by the tissue, the cut off frequency of the measurement system is extremely high, such that the measurements comfortably accommodate the MTF with its long tail. This statement was intended to convey the observation of a long-tailed MTF of the retina with non-zero residual contrast (a characteristic of MTF in scattering-dominated systems). We now have modified the statement for clarity:

“In contrast to many lens-based optical systems, retinal MTFs have a long tail with non-zero residual contrast despite an initial rapid loss of contrast (a characteristic of scattering-dominated optical systems). The monotonic decay of retina-transmitted contrast indicates scattering-induced veil rather than a frequency cut-off to be the cause of contrast loss…”

The use of limited aperture in the system means that it will have its own MTF. The authors should show the optical system MTF in their plots on Figure 3.

Again, this is a very good and insightful point. Although not very prominently addressed, this data was already present in the first submission (see manuscript Figure 3—figure supplement 1C). The MTF of the optical set up is now included in a separate panel in Figure 3—figure supplement 2H. The authors feel that inclusion of the MTF measurement of the optical set up (which has already been accounted for in the retinal MTF calculations, kindly see Materials and methods) in the main Figure 3 would be a technical detail that does not directly impact the interpretation of the biological message conveyed. The overall optical quality of the microscope set up is an order of magnitude better than that of the retinae.

9) In the same subsection: The initials T.V. should be deleted.

We apologise for the oversight. This was a citation that was not properly included. It has been fixed now.

10): What range of spatial frequencies were used for these computations?

The range of spatial frequencies used is 0-2 cycles/deg. The relevant lines in the revised manuscript have been modified to include this information:

“With regards to our MTF measurements, we that find the Strehl ratio (computed using measurements in the spatial frequency range of 0-2 cycles/deg) of a fully developed retina is increased 2.00 ± 0.15-fold compared to that of pups (P14) in which chromocenter fusion was not completed, and similarly 1.91 ± 0.14-fold (ratio of means ± SEM) improved compared to TG-LBR adult retinae (p = 3.4055e-08) in which chromocenter fusion was deliberately arrested (Figure. 3D).”

Please also refer to earlier section “Complete MTF representation" in response to the editor for an explanation that justifies the depiction and complete measurement of MTF for frequencies beyond the behaviourally relevant regime.

11) Subsection “Improved retinal contrast transmission”, Figure 3:D2 and D3, subsection “PSF measurements”: The intensity of the PSF in the figure is lower for the TG-LBR mouse across the entire displayed range of -20 to 20 microns. But the authors state that the integrated intensity is the same between the two when the PSF is integrated over an 80 x 80 micron area. I am very skeptical that the integrated intensity under the two curves in Figure 3:D2 will become equal.

Thank you for raising the concern. We agree that it feels counterintuitive that the intensities in the displayed ROI for the TG-LBR result in the same integral intensities in both cases. The ambiguity stemmed from the different sizes of ROI chosen for display and the ROI chosen for the normalization of the PSF curves to ensure same total integral intensities. Normalization was done over a window of 80x80 μm, whereas we show only the central 40x40 μm of the image. The description is therefore modified for clarity and reads as follows in the revised manuscript:

“PSF measurements

The point spread function (PSF) measurements were carried out using a 40 μm pinhole (P40H, Thorlabs) acting as a point light source, such that the demagnified point projected on the retina was of the size about 3 μm. Raw images were corrected for background by subtraction of a dark frame in FIJI. Resulting images were normalized with respect to the integral intensity in the field of view (~80 μm x ~80 μm), and the central region with an ROI of 40 μm by 40 μm was cropped, averaged and displayed in false color.”

Furthermore, please find in Author response image 1 a representation of the PSF in a larger field of view. When going beyond the 20 μm ROI of the PSF intensity images, the local intensities of the TG-LBR retinal PSF is equal or higher than the WT retinal PSF (inset in Author response image 1). Given the higher radial weight, this makes the integrated intensities converge to the same value. To further clarify this aspect, the ensquared energy of the PSFs have been provided for an ROI ~525 μm x ~525 μm, showcasing the total intensities in both retinal types asymptotes to the same value. Notably, at a center-to-edge distance of ~40 μm (ROI of ~80 μm x ~80 μm, which we choose to prevent the collection of too much dark current), the integrated intensities reach ~97% and ~93% of the final values in WT and TG-LBR retina, respectively. Thus, our earlier normalization in the smaller ROI does not affect the Strehl Ratio measures by more than 4%. Notably, if we would re-normalize the data to take into account the full field of view (but potentially including also more camera noise), the difference between the peaks would even be higher, as the TG-LBR peak would be 4% lower. For reference, the results of the encircled energy have been included in the supplementary information of the revised manuscript (Figure 3—figure supplement 2I).

**Author response image 1. respfig1:** Point spread function of the retinae. PSF intensities of retina transmitted point source images. Inset showing region where the intensities of the TG-LBR retina is higher than that of WT retina such the total normalized intensities in both cases is the same.

12) Results section final paragraph: "This suggest…."

Sorry, we corrected this.

13) Discussion paragraph one: The lack of `nuclear inversion' in diurnal animals is intriguing and the authors make a very sensible suggestion that the ONL is significantly thinner in diurnal animals. However, that statement should be backed up by proper citations or, better yet, a table or a plot comparing ONL between nocturnal and diurnal animals.

We agree that this information was poorly accessible from this section. References appeared for a similar statement in the introduction. They have been re-added here with the following additional references:

1) Solovei, I., Kreysing, M., Lanctôt, C., Kösem, S., Peichl, L., Cremer, T., Guck, J., and Joffe, B. (2009). Nuclear architecture of rod photoreceptor cells adapts to vision in mammalian evolution. Cell 137,356–368.

2) Werner, J.S., and Chalupa, L.M. (2004). The Visual Neurosciences (MIT Press).

3) Williams, R.W., LM, S.M., JS, W., 2003 (2003). Developmental and genetic control of cell number in the retina.

4) Sterling, P., and Laughlin, S. (2015). Principles of Neural Design (MIT Press).

“Wouldn’t improvements in retinal image contrast not also be beneficial for diurnal mammals? Firstly, the larger spacing of photoreceptor segments in the diurnal retina significantly reduces ONL thickness (Solovei et al., 2009, Sterling and Laughlin, 2015; Werner and Chalupa, 2004; Williams and Moody, 2003) and thereby the risk of scattering induced veil and loss of image contrast.”

14) Materials and methods: In general, this section is sloppily written with numerous typos, combinations of present and past tense – often in the same sentence – and unclear writing. There are numerous typos. The authors flip between the abbreviation SR and strehl ratio.

We apologise for numerous typographical errors in this section and thank the reviewer for pointing them out. These have been rectified. A summary of the changes in the Materials and methods section has been provided in response to editor point 1.

15) Subsection “Calculation of MTF”. How do the authors propose to use this technique to measure optical impact of outer segments? Note that ex vivo preparations are vulnerable to optical artifacts, especially the delicate optical properties of the retina.

This is a good point. We think we were not completely clear in our descriptions here. What we meant to say in the relevant part of the Materials and methods section is that genetically changing the optics of the ONL enables a better understanding where scattering occurs in the retina. This can also be compared to the performance of the lens, as reported by other studies. For optical contributions of the outer segments we mostly referred to the literature, and we would like to clarify that it is not straight forward to experimentally confirm the contribution of the segment layers by our method, and we don’t see highest need for it as our experiments include the direct comparison of 2 retina phenotypes with outer segments present.

To further investigate the relative contribution of inner retina vs segment layers, simulations were used, but we only really touched on these aspects. We present in Author response image 2 more details of our simulations. The literature suggests that the outer segments have limited impact on MTF (they have wave guiding properties, (Vohnsen, 2007; 2014)). Gaining optical access to pure OS structures is admittedly is not trivial. Therefore, we adopted the simulation strategy to exclusively look at light wave propagation in the OS, which gave similar results to those of previous studies (Vohnsen, 2007; 2014) (Author response image 2).

**Author response image 2. respfig2:** Modelling light propagation in mouse rod OS. (Top panels) RI distribution in the rod OS assumed at a contrast of 6-8% and OS cylinder diameter assumed 1.6um and 25um long. (Bottom) Intensity distribution of a plane wave propagating longitudinally in the cylindrical outer segment structures indicating a guiding effect. Simulations performed using Biobeam (Weigert et al., 2018).

These simulations were then used to present a conservative estimate of the impact on the MTF according to previous models and show that it is low (Figure 3—figure supplement 2G). Results show that outer segments only have a negligible impact on the overall MTFs (Figure 3—figure supplement 2G), in agreement with previous experimental findings (Enoch, 1963) and models of the outer segment (Vohnsen, 2007; 2014) acting as waveguides.

Moreover, this issue applies to a young mouse, where outer segments are not fully developed. If the outer segments were to have an impact, with the growth of the outer segments, the loss of contrast would be only more rapid. However, we show an improvement in the optical properties of the retina as it develops. Furthermore, we also show the same response in the adult transgenic retinae in comparison to the retina of a mouse pup. This also implies that only ONL has an optical impact. Together, these findings gave us confidence that the fragile outer segments did not bias our results. To ensure that a misleading idea is not conveyed to the readers that the effect of the fragile outer segments could be assessed with the optical set up, the relevant lines in the Materials and methods have been modified as follows:

“The differential readout of the transmitted image through the inversion arrested TG-LBR retina allows an explicit understanding of the optical impact of the inner retina and the outer nuclear layer architecture in relation to other ocular constituents, such as the lens and the reported optical properties of mouse eye in *in-vivo* studies (Geng et al., 2011; la Cera et al., 2006; van Oterendorp et al., 2011). As for the photoreceptors outer segments, their impact is minimal as they act as waveguides as described in previous ex vivo studies (Ohzu et al., 1972). Such an effect is also verified by our simulations.”

16) Behavioral assessment: What does the temporal frequency mean? Was the stimulus flickering? Or moving, or both? This entire section is very poorly written.

We are grateful to the reviewer for raising this concern and we apologise for any ambiguity in the description. The stimulus was not flickering (meaning there were no temporal changes in light intensity) The description of the method has been updated for clarity:

“The temporal frequency here refers to the combination of spatial frequency (cyc/deg) and speed of movement in (deg/s), which gives an effective temporal frequency, namely the change of contrast at a given point on the screen which was maintained constant at a particular temporal frequency (0.73 cyc/s or Hz).”

Also, kindly refer to our response to the editors point 1above for a revised version of the behavioural experimental methods.

17) Subsection “Image processing and segmentation of ONL model”: Why was this smoothing necessary? Were the final results different when they were not smoothed? Does the smoothing generate refractive index profiles that are more realistic?

As the reviewer correctly assumed, the reason we applied some smoothing was to remove sharp transitions that we considered to be biologically less realistic. Also, sharp boundaries frequently may increase simulation artefacts. But we did not represent in detail how much this smoothing would change results. To address the effect of smoothing, we reran our simulations without any additional blurring step. Although the overall scattering cross section did slightly change, as expected (since the refractive index map changes), the relative reduction in side scattering for the inverted case compared to the conventional architecture was even found to be even slightly increased (65% vs 58%). Hence the smoothing leads to conservative results, and the gain of nuclear inversion might be even slightly stronger than inferred by our simulations. See Author response figure 3.

**Author response image 3. respfig3:** Differential simulations of light propagation in the ONL, illustrating differences between the use of a refractive index maps with (**A**) and without (**B**) blurring of the refractive index distributions.

18) Subsection “Relative contributions to MTFs from ONL and outer segments”: Replace OS with 'outer segment'

Thank you. This has been fixed.

Perhaps the Matlab script mentioned in the text should be shared.

The MATLAB script that we used has been cited and can be obtained from the link within this publication. The Materials and methods section has also been updated with parameters used for calculations:

“Mie calculations were implemented via a modified MATLAB script (Mätzler, 2002) that can be downloaded at the following link – https://omlc.org/software/mie/. The relevant parameters used are m_euchromatin/medium=1.02, m_heterochromatin=1.04, which are refractive index of the euchromatin/medium and heterochromatin/particles, respectively. The wavelength used was 500nm and volume fraction vf = 0.3351. The diameter of particles used was in the range 0.92-2 𝜇m.”

Reviewer #2:Paper Summary:The authors build on a body of literature that has identified the interesting phenomenon of "nuclear inversion" in nocturnal mammals. In this report, the authors test the hypothesis that the re-organization of euchromatin and heterochromatin within the nucleus of rod photoreceptor cells could serve to benefit nocturnal mammals by reducing scatter in the outer nuclear layer which is thick in rod-dominant mammals such as mice. An impressive set of data is collected in the report. The authors interpret their findings as supportive of a role of improved contrast sensitivity due to nuclear inversion which purportedly reduces optical scatter, and thereby improves the contrast ratio of images that must project through all retinal layers before striking the outer segments of rods.The paper is thoughtfully composed and was generally a pleasure to read. The data set is impressive and authors are congratulated on a wholesome battery of tests that span in vitro preparation, phantom simulations, mouse behavioral testing, histology with immunolabeling and transgenic animals that support the general hypothesis.

Thank you for summarising the research findings, acknowledging our motivation to study the optics of the retina and appreciating our interdisciplinary approach to address the long-standing hypothesis of the role of nuclear inversion in nocturnal mammals.

The major criticism for the report, however questions the very raison d'etre of the manuscript; "just how beneficial is this nuclear inversion to mouse visual performance?" While nuclear inversion is indeed a strange behavior of outer retinal cells (especially rods), it is unclear whether this is an epiphenomenon of some other function important to rods, or whether, as the authors would suggest, truly provides visual contrast benefit to the animal. The authors provide some evidence in support of this idea, but there are several misleading conclusions drawn from figures (especially Figure 3) which overstate the contrast benefit to mice by using simulations that are not behaviorally relevant.

We appreciate the points raised by the reviewer. We agree that particularly in manuscript Figure 3 we probed spatial frequencies beyond the acuity limit of mice. We agree with the reviewer that our reasoning for this was not very well communicated. Below, we provide a more detailed explanation in cases where we still think it is justified to extend the optical analyses to these high frequencies, or we have updated measurements and additional panels showing the results for visually relevant regime. We furthermore explain why we think that the main conclusion of the manuscript remains valid and apologize for a missing part of a figure (manuscript Figure 4F) caption that might have made the interpretation of the data difficult.

Problem 1: Authors show the MTF improvement of contrast transmission when projecting sinusoidal patterns directly onto the retina. The differences in retinal contrast appear impressive in Figure 3AB. When comparing pups or TG-LBR mice (which also do not have nuclear inversion) to the adult WT mice that do have nuclear inversion, contrast transmission appears to increase. However the range of spatial frequencies tested are not generally thought to be behaviorally relevant to mice. Reports by Histed MH, Carvalho LA, Maunsell JH. (J Neurophysiol 2012, and corroborated by a multitude of other studies) suggest that maximum spatial frequency cutoff for the mouse is near 0.5 cyc/deg. This represents the very lowest of the tested spectrum in Figure 3AB. By those measures, roughly 2/3 of the data is behaviorally irrelevant to the normal mouse. When considering data from 0-0.5 cycles/degree, the effect is visually modest in comparison. Reviewer requests revision of the figure to reflect the improvement range to that closer of what is relevant to mouse visual behavior.

Thank you for the remark. We are aware that some of the frequencies that we probed go beyond the behaviourally relevant regime, particularly under low light conditions. We further acknowledge that presentation of data in the original submission was not well suited (log-scale) to access the visual benefits in the behaviour relevant regime. However, data presented as an integrated optical quality measure in the relevant spatial frequencies (0 – 0.3 cycles per degree) had already been provided in manuscript Figure 3—figure supplement 2D. We appreciate the reviewer’s point about showing more behaviourally relevant data here, and we have therefore added a panel illustrating the contrast-transmission characteristics for behaviourally relevant spatial frequencies. Now, manuscript Figure 3(C) depicts the MTF transmission in the visually relevant spatial frequencies. Also, we made the data more accessible by using a linear, rather than logarithmic scale. This data clearly shows that in the relevant frequency regime, contrast is increased by 33-45% which is similar in magnitude to the behavioural benefit demonstrated in terms of contrast sensitivity improvements which is in the range 18-27%. The experiments with the approach of a predator by projection of a cat face have now been updated for the visually relevant distances. Kindly refer to the sections “Complete MTF representation” and “Cat Image**”** for further detailed answer to these concerns.

Given the interdisciplinary nature of the work and the broad audience, we would like to keep the other MTF and psf figure representations intact. The results demonstrating a 2-fold increase in optical image transmission capabilities are important from an optical perspective (Strehl ratio is likely the best-established metric for optical performance) and furthermore reveal the magnitude of optical penetration that could be achieved in living tissues, if their optical properties could be controlled (our future research direction). As such, we believe that these data are of interest to the wider biological microscopy community and should be retained in the manuscript.

Problem 2: Projection of 3 micrometer PSF into the mouse retina (Figure 3D) is behaviorally irrelevant. Based on the literature that the authors cite (and more), Geng et al., Schmucker and Schaeffel 2004 and others such as Remtulla and Hallett (1985), a 3 micron PSF is a highly unnatural stimulus for the mouse retina because of spherical aberration, longitudinal chromatic aberration, transverse chromatic aberration, a constantly growing mouse eye and an optically thick retina. Anything less than a single-wavelength stimulus therefore would be impossible to naturally project at a 3 micrometer spot, and thus it is unclear why the authors are using this highly unnatural stimulus to model the PSF spread in Figure 3D.

We discussed this in detail in response to the editors’ questions (see section PSF measurements). In brief: The PSF measurements were used as an independent confirmation of the MTF measurements (especially their volumes). Additionally, we think it is interesting to characterize the point spread performance of the retina, a heterogeneous biological tissue, as an independent optical element of the eye. The PSF measurements also complement the frequency domain MTF measurements. At this juncture, we would like to clarify the superiority of the frequency domain MTF analysis to pin point the improvements imparted by the retina in visual behaviourally relevant spatial frequencies. To reiterate and clarify the quantification of the “point spread”, we agree that the point stimulus as projected by the ocular system of the mouse will indeed be aberrated. However, the PSF spread shown in the manuscript Figure 3 does not relate to the PSF of the ocular system of the mouse as projected on to the retina. The PSF measurement serves as an independent confirmation of the improved optical quality according to canonical measure ‘Strehl ratio’. We did not intend to convey that such a point source is an ecologically relevant stimuli, either with or without aberrations by the lens. These points have been made clear in the revised manuscript. For more details kindly refer to the section on PSF measurementsin response to the editors, which also justifies the presentation of individual non-averaged point source.

Problem 3: Authors attempt to simulate the behavioral benefit to the mouse by a friendly example of what the mouse would "see" in an approaching cat by showing a phantom of the cat face. This is a fun example, but again represents a scenario that is unlikely due to the visual acuity of the mouse (adult or otherwise). If assumed that behavioral spatial frequency is limited to ~0.5 cyc/deg, there is little chance the mouse would visualize the cat eyes at any distance represented by Figure 3. The reviewer calculates that interpupillary distance of a typical house cat (which is assumed to be a biotypical natural predator of the mouse? certainly not a tiger!) is 36 mm (following Hughes, 1972 Vision Research). If we are generous and round this to 4cm, the subtended angle on the mouse retina will surely not render the eyes of the cat in such a way that the authors illustrate. At 4 meters, subtended angle is nearly 0.57degrees. At 2 meters, subtended angle is 1.14 degrees. Again, this far exceeds the reported visual acuity of the mouse and therefore the example is inappropriate, behaviorally irrelevant and is misleading to the general scientific audience. There would be no visual benefit to the mouse in these conditions even if nuclear inversion were found to benefit contrast transmission. Request removal of this figure.

This is a valid point that was also raised by other reviewers. Thank you for the detailed calculations pointing out that the illustration may not be relevant to mouse visual behaviour. The experiments with the approach of a predator by projection of a cat face have now been updated to show visually relevant distances. In brief, visual acuity only determines the resolvability of features. This does not provide information about the presence or absence of a stimulus. We however have taken the reviewers’ valid concerns into account and repeated the experiments of the approach of a cat for visually relevant distances. New results show a comparable advantage at distance below 1 meter (70cm, 45cm and 25cm). The figure has been revised accordingly. Kindly refer to the section on “Cat Image”in response to the editor for more details.

Problem 4: Problems 1-3 are further compounded that the generous spatial frequency cutoff for the mouse is 0.5 cycles/deg for photopic conditions (Prusky et al., 2000; Histed et al. 2012). Spatial frequency tuning for the WT mouse is considerably worse under scotopic conditions which is the regime that stands to benefit from rod nuclear inversion (authors report this is a rod-dominated effect and cones generally do not show such behavior). Umino, Solessio and Barlow, 2008, show scotopic contrast sensitivity is even lower than photopic in the mouse. Behaviorally tested cutoff is near 0.2 cyc/. When this is projected back on to the data from Figure 3AB,D1,D2,E and F) the behavioral benefit in Figure 3 seem to be baseless.

Again, referring to sections Complete MTF representationt, he authors would like to point out that the differences in the contrast transmission by the retina at ~0.28cycles/degree is around 33-45% which is similar in magnitude to the behavioural benefit demonstrated in terms of contrast sensitivity improvements which is in the range 18-27%. We hope this substantiates the causal relationship between the retinal contrast transmission and the differences in the behavioural contrast sensitivity.

For the cat image, please refer to our answers above. With respect to the individual point sources, we think it is worth showing them because i) they complement intensity-averaged PSF measurement, and make the amount of fluctuations better accessible, ii) we would like to point out that detectability and resolvability are not the same. As in microscopy, the inability to resolve a point source like fluorophore, does not mean that it cannot be detected. iii) adding to this, it is a rather well-known phenomenon that intensity distribution may impact detectability (which explains stars can be better seen with glasses, although they remain unresolvable to the eye), iv) more recently the strong evidence for non-linear / thresholded detection has been demonstrated for the human eye (Tinsley et al., 2016).

Despite these shortcomings, the manuscript has merit. Problems 1-4 are somewhat mitigated by compelling data in Figure 4 which do show a slight benefit in WT mice (with nuclear inversion) vs LBR mice which presumably do not. Scientific audience is left to trust that TG-LBR mice have otherwise normal ocular behavior with the exception of high-chromocenter rod nuclei. Further description of the phenotype would convince skeptics further (including eye size and anterior optical media clarity which could also account for the result in Figure 4).

Thank you for these suggestions.

Before addressing the concerns of the reviewer, we wish to highlight and expand upon the already existing control experiments in the manuscript providing some evidence that the TG-LBR mice have otherwise normal visual behaviour except for their arrested nuclear inversion in rod photoreceptors.

a) There is no difference in diurnal vision (manuscript Figure 4B, Left).

b) Similar sensitivities in the lowest spatial frequencies (manuscript Figure 4B, Right, manuscript Figure 4D) where differential retinal-contrast transmission was also comparable (manuscript Figure 3B, 3C) indicate an uncompromised downstream neural visual pathway that is responsible to low spatial frequency roll off of the CSF.

c) Comparable absolute transmission in the retina (manuscript Figure E3) indicates the absence of any non-physiological state that could lead to protein aggregates inducing isotropic scattering and alter photon transmission.

Furthermore, we now provide experimental data comparing the ocular parameters of the WT and TG-LBR mice. The axial lengths of the eye and the diameter of the lens are not significantly different between the two phenotypes. This further confirms that the differences in contrast sensitivities close to the visual acuity of mice arises solely from the differences in retinal-contrast transmission. These measurements are now reported in Figure 4—figure supplement 2.

Nevertheless, the complexity of the visual pathway is too high to rule out all thinkable side effects. For instance, a 30% down-regulation of a sparse neuro-receptor that even deep sequencing would have little chance to pick up, could also be affecting visual behaviour. Indeed, the question remains whether the modest contrast loss in the retina is sufficient to explain the approx. 20% improvement in contrast sensitivity. In the existing manuscript, however, we already provide evidence that the Strehl ratio improvements in region of sensitivity (0 – 0.36 cycles / degree) is 24% ± 14%, which is similar in magnitude to the integrated sensitivity improvements seen in behaviour (18-27%). Specifically, at 0.28 cylces / degree where behavioural benefits are strongest, nuclear inversion improves the optical quality of the retina 33%-45%. But vision remains a highly non-linear process, and the reviewer is right that knowledge about differential contrast transmission alone might not be sufficient to explain the differential sensitivity.

How can we then show than that the retinal contrast loss is sufficient to account for the loss of sensitivity under low light conditions? The answer is a recueexperiment. (see also our response to point 3 raised by the editor). Perturbations often come with a risk of unspecific side effects. To relate an observed phenotype and a protein hypothesised to be involved in a particular cellular pathway, a candidate protein may be knocked down via RNAi interference. To show specificity, a recue should be performed by interfering with the pathway as close to the final phenotypic read out as possible. Our optical rescue experiment is analogous to a recue in an RNAi interference experiment. We did not stop by showing that sensitivity is increased, as the reviewer incorrectly stated. In addition to showing an enhancement in contrast sensitivity, we designed and performed such a rescue experiment to explicitly show that the contrast loss in the retina is sufficient to account for the loss of sensitivity.

As explained in the text (see paragraph beginning “Finally, we asked whether reduced visual sensitivity of mice lacking the inverted nuclear architecture can be sufficiently explained by inferior contrast transmission of the retina”), we designed the rescue experiment such that we would restore only the optical consequence of the TG-LBR phenotype. This is included in the last panel (F) of manuscript Figure 4 and Figure 4—figure supplement 2A-C, and although described in detail in the manuscript, unfortunately in the submitted version, the figure legend got cropped. We apologise for the inadvertent omission. We wish to thus elaborate on our motivation to perform these experiments.

First, the linearity of optical systems dictates that contrast losses are always relative, and independent of absolute contrasts. Because this is the case, we can describe the optical part of the vision systems as a serial system of contrast modulators, starting with the lens, and ending just before the light sensitive segments. This linear description implies that we can compensate the optical consequences of inversion arrest in rod nuclei. That is, we can pre-compensate the contrast such that photoreceptor outer segments of the TG-LBR mouse experience the same contrast level as in the WT retina. When we did this experiment, we found that under these conditions visual sensitivity was restored (manuscript Figure 4 F).

The beauty of this rescue furthermore is that it is highly specific, because very far downstream in the visual pathway, and as such does not bear significant risks of restoring the retina from LBR transgene side effects. Clearly, we cannot rule out other side effects, for which however we observed no evidence at level of retinal (Figure 2—figure supplement 1), ocular or lens anatomy (Figure 4—figure supplement 2), and non-limiting rod vision was normal (Figure 4B (Left)). Our rescue experiments clearly show that the extra loss of retinal contrast transmission downstream of LBR expression and inversion arrest is sufficient to explain the reduced sensitivity in rod inversion arrested mice. We hope that these explanations bring clarity to the relevant description of the rescue experiments in the manuscript that was unfortunately not assessed by the reviewers.

Given these arguments, we maintain our previous conclusion that a gain in contrast transmission is sufficient to explain the gain in sensitivity. To make sure this important point is properly conveyed, we extend the relevant part in the paper. At the same time, the reviewers are right that we cannot fully exclude that there are also other functions of nuclear inversion. (see also our response to point 3 raised by the editor).

In the discussion, the authors do not provide enough latitude that other epiphenomenon and bioselection-driven reasons for nuclear inversion are possible. The manuscript would be stronger if such openings for these possibilities are explored further. The reader is left with the feeling that the problem is solved, which it is not. Data is provided to support a hypothesis.

This is likely the only point where we disagree with any of the reviewers. We refute the view that nuclear inversion, and if we understand the reviewer correctly also the reduced retinal light scattering, could be an epiphenomenon that does not explain the differential sensitivity of mice. We believe that we provide strong, unambiguous evidence for a causal relationship with the occurrence of nuclear inversion. Most centrally, as we explained above, the rescue experiment shows that improved retinal contrast transmission is sufficient to explain improved contrast sensitivity. However, we agree that we could have been more through in explaining how we arrived at our conclusion and openly discuss alternative explanations. We have accordingly addressed open questions and potential additional benefits of rod photoreceptor nuclear inversion. Please refer to our response to the editor Point (3).

Figure 4F not described in Figure 4 caption.

Thank you. We apologize for the inadvertent omission of the figure caption. We are sorry that this information was not accessible and might have made it difficult to follow our sufficiency argument. This has been rectified. The corresponding caption now reads as follows:

“(F)Rescue experiment demonstrating sufficiency of improved retinal contrast transmission to explain improved sensitivity. Adjusting the level of contrast at the photoreceptor level (by pre-compensation of differential contrast loss) restores sensitivity of TG-LBR mice. N indicates number of individual trials of 10 animals together for each mouse type.”